# Adversarially Robust 3D Point Cloud Recognition Using Self-Supervisions

**Jiachen Sun** [*1], **Yulong Cao** [1], **Christopher Choy** [2], **Zhiding Yu** [2], **Anima Anandkumar** [2,3], **Z. Morley Mao** [1], **and Chaowei Xiao** [2,4]

[1] University of Michigan, [2] NVIDIA, [3] Caltech, [4] ASU

## Abstract

3D point cloud data is increasingly used in safety-critical applications such as autonomous driving. Thus, the robustness of 3D deep learning models against adversarial attacks becomes a major consideration. In this paper, we systematically study the impact of various self-supervised learning proxy tasks on different architectures and threat models for 3D point clouds with adversarial training. Specifically, we study MLP-based (PointNet), convolution-based (DGCNN), and transformer-based (PCT) 3D architectures. Through extensive experimentation, we demonstrate that appropriate applications of self-supervision can significantly enhance the robustness in 3D point cloud recognition, achieving considerable improvements compared to the standard adversarial training baseline. Our analysis reveals that *local* feature learning is desirable for adversarial robustness in point clouds since it limits the adversarial propagation between the point-level input perturbations and the model's final output. This insight also explains the success of DGCNN and the jigsaw proxy task in achieving stronger 3D adversarial robustness.

## 1 Introduction

Point cloud data is one of the most broadly used representations in 3D computer vision. It is a versatile data format available from various sensors and computer-aided design (CAD) models. Given such advantages, many deep learning-based 3D perception systems have been proposed [1–6] and achieved great success in safety-critical applications (*e.g.,* autonomous driving) [7–9]. Although deep learning [5, 10] on point clouds has exhibited high performance, they are particularly vulnerable to adversarial attacks [11–13]. Because of the wide applications in safety-critical fields, it is imperative to study the adversarial robustness of point cloud recognition models.

However, there are at present two obstacles on the path reaching robust point cloud recognition:

**Architectural Diversity.** Deep 3D point cloud recognition is an emerging field. Many 3D architectures with various feature aggregation methods have been proposed. We broadly categorize them into three families based on how networks aggregate the geometric information: multi-layer-perceptron (MLP)-based networks [5, 10], convolutional networks [3, 4, 6, 14–16], and transformer-based networks [17, 18]. These networks have been studied mostly using clean accuracy as their main metric, but an in-depth study of their robustness is lacking.

**Vulnerability to Adaptive Attacks.** A few defenses against 3D attacks have been recently proposed [19–21]. Some of the methods, however, merely obfuscate attackers by limiting the malicious agents from accessing the defense systems and true gradients [22]. They have been shown vulnerable to adaptive attackers who have the full knowledge of the defenses and can approximate the gradients [21]. Similar to the 2D domain [23], adversarial training (AT) [21] provides more longstanding robustness in 3D even against adaptive attacks [22]. Nevertheless, further robustness improvements on adversarial training are highly desired for practical usage and deployments.

---

[*]Correspondence to `jiachens@umich.edu`

35th Conference on Neural Information Processing Systems (NeurIPS 2021)

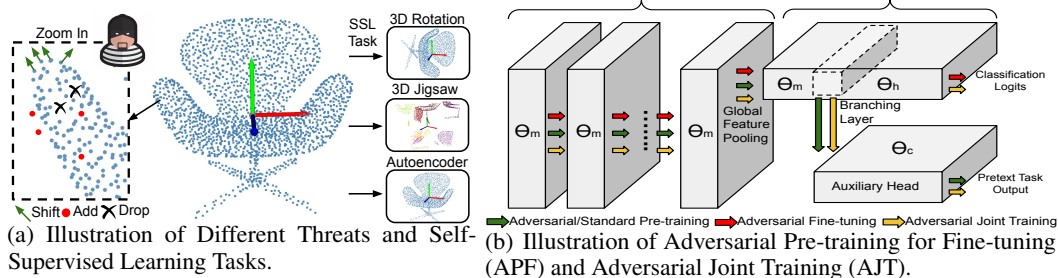

(a) Illustration of Different Threats and Self-Supervised Learning Tasks.

(b) Illustration of Adversarial Pre-training for Fine-tuning (APF) and Adversarial Joint Training (AJT).

Figure 1: Overview of Our Analysis in 3D Point Cloud Classification.

Self-supervised learning (SSL) has been incorporated into adversarial training in 2D image perception models lately. It has shown great potential to enhance adversarial robustness without requiring any additional data or labels [24, 25]. Given such achievements, a natural question emerges: can we mimic the application of SSL to improve adversarial robustness in 3D point cloud recognition? Such a label-free strategy is preferred due to the cost and difficulty of 3D point cloud data annotation [26].

**Summary of Our Contributions**:

In this paper, we present a systematic analysis of the adversarial robustness in 3D point cloud recognition using self-supervisions on three representative architectures: a multi-layer-perceptron (MLP) network (PointNet) [5], a convolutional network (DGCNN) [15], and a transformer-based network (PCT) [17]. Specifically, we use two strategies to integrate self-supervised learning and adversarial training, including (1) adversarial pre-training for fine-tuning (APF), which uses the SSL tasks only for pre-training, and (2) adversarial joint training (AJT), which jointly trains the SSL task with the recognition task, as shown in Figure 1. To further study the importance of self-supervised tasks for adversarial robustness, we select three representative SSL proxy tasks, including 3D rotation prediction [27], 3D jigsaw [28], and autoencoding [29]. Our key observations are as follows:

- We show that pre-training on SSL tasks improves adversarial robustness of the fine-tuned models. Unlike the 2D domain, where both APF and AJT have enhanced the robustness, our study finds that only APF consistently achieves robustness improvements in 3D. AJT does not always help since the distributional gap between data for SSL and recognition tasks will distract each other in AJT. Evaluation results of various unforeseen attacks further confirm such improvements by APF.

- We find that the convolutional network, *i.e.,* DGCNN, is more robust than the other architectures in point cloud recognition tasks. Moreover, 3D jigsaw SSL task, which predicts the permutation of 3D point cloud patches, helps achieve stronger robustness than the others. Both convolutional architecture and jigsaw SSL task enforce the model to learn better local semantics. Intuitively, robust local features help limit the propagation of adversarial effect from point-level input perturbations to the model's final output.

- We demonstrate that fine-tuned models from different pre-training tasks have different vulnerabilities, and adversarial examples generated by attacking them do not transfer well among each other. Thus, we further leverage two simple yet powerful ensemble methods to boost the adversarial robustness by a substantial margin. Our best ensemble models, for instance, achieve robust accuracy of 53.5% (**+15.6%**), 69.4% (**+7.4%**) and 57.9% (**+8.8%**) with PointNet, DGCNN, and PCT on the representative dataset, ModelNet40 [30].

## 2 Analysis Methodology

In this section, we detail our adversarial robustness analysis methodology. We first introduce the principal 3D point cloud recognition architectures and the threat models used in our study. We then introduce two ways to generalize and improve AT using 3D point cloud SSL proxies.

### 2.1 3D Point Cloud Recognition Models and Threats

We introduce the adopted model designs and the formulations of threats to 3D point clouds below.

**Model Variants.** We use a shared multi-layer-perceptron-based network PointNet [5], a convolutional network Dynamic Graph CNN (DGCNN) [15], and a transformer-based network Point Cloud Transformer (PCT) [17] as our primary backbone architectures, denoted as $\mathcal{M}_{\boldsymbol{\theta}_m}$. Specifically, PointNet directly aggregates learned features from each point to form a global embedding for final

recognition. By doing so, the point-level features are independent of each other before global pooling. PointNet embraces high efficiency that has been widely utilized as a base operation in complex tasks [8, 31]. DGCNN instead builds a graph based on $k$-nearest neighbors ($k$NN) and uses a variant of continuous convolution (EdgeConv) on edges to enable local feature learning. The combination of local and global representation learning helps DGCNN achieve a higher clean accuracy. PCT extends the Transformer [32], the dominant framework in natural language processing, for point cloud recognition. The core of PCT is the full attention mechanism, which establishes a much more flexible scheme, where each point has the potential to affect every other point in the point cloud. The classification head $\mathcal{H}_{\boldsymbol{\theta}_h}$ parameterized with $\boldsymbol{\theta}_h$ for these backbones is an MLP and the part segmentation head $\mathcal{H}_{\boldsymbol{\theta}_h}$ is a set of $1 \times 1$ convolutions. We use $\mathcal{F}_{\boldsymbol{\theta}_f}$ parameterized with $\boldsymbol{\theta}_f$ ($\boldsymbol{\theta}_f :=$ $[\boldsymbol{\theta}_m; \boldsymbol{\theta}_h]$) to represent the overall model architecture, consisting of the stacked backbone $\mathcal{M}$ and recognition head $\mathcal{H}$, where $\mathcal{F} = \mathcal{H} \circ \mathcal{M}$. Given the input point cloud $\boldsymbol{x}$, the model $\mathcal{F}$ aims to predict the corresponding label $\boldsymbol{y}$, where $\boldsymbol{y} = \mathcal{F}(\boldsymbol{x})$. More details of the architectures are in the supplements.

**Threats**. There are mainly three types of threats against point cloud perception models, which could be abstracted as point shifting (PS), point dropping (PD), and point adding (PA) attacks (Figure 1(a)). We formally define these threats within $\ell_p$ projected gradient descent (PGD) style attacks. First, we assume a PS adversary is able to shift all existing points within a $\ell_p$ norm ball:

$$\boldsymbol{x}_{s+1} = \Pi_{\boldsymbol{x}+\mathcal{S}}(\boldsymbol{x}_s + \alpha \cdot \text{sign}(\nabla_{\boldsymbol{x}_s}\mathcal{L}(\boldsymbol{x}_s, \boldsymbol{y}; \mathcal{F}))); \quad \boldsymbol{x}_0 = \boldsymbol{x} + \text{U}(-\epsilon, \epsilon) \tag{1}$$

where $\boldsymbol{x}_s$ is the adversarial example in the $s$-th iteration, $\Pi$ is the projection function to project the adversarial example to the pre-defined perturbation space $\mathcal{S}$, $\alpha$ is the attack step size, and $\text{U}(-\epsilon, \epsilon)$ represents a uniform distribution from $-\epsilon$ to $\epsilon$. Second, we allow a PD adversary to discard the top $k$ salient points [33] in each iteration until a total drop of $N$ points:

$$\boldsymbol{x}_{s+1} = \boldsymbol{x}_s \ / \ \arg \text{top}k_{\boldsymbol{x} \in \boldsymbol{x}_s} \text{saliency}(\boldsymbol{x}_s, \boldsymbol{y}; \mathcal{F}); \quad \text{if} \ \ k \times s < N \tag{2}$$

where we follow [33] to implement $\text{saliency}()$ to calculate the importance scores of the input points *w.r.t.* to the prediction accuracy and $/$ denotes the dropping operation. We detail $\text{saliency}()$ in the supplements due to space limits. Third, a PA adversary is capable of adding new points bounded by an $\ell_p$ norm ball to the original point cloud. PA randomly initializes $N$ new points from the original point coordinates and only perturbs the added points. We follow the same setting as PS to formulate the perturbation:

$$\boldsymbol{x}_{s+1} = \Pi_{\boldsymbol{x}+\mathcal{S}}(\boldsymbol{x}_s + \alpha \cdot \text{sign}(\nabla_{\boldsymbol{x}_s}\mathcal{L}([\boldsymbol{x}_{ori}; \boldsymbol{x}_s], \boldsymbol{y}; \mathcal{F}))); \ \boldsymbol{x}_0 = \text{sample}(\boldsymbol{x}_{ori}, N) + \text{U}(-\epsilon, \epsilon) \tag{3}$$

where $\boldsymbol{x}_{ori}$ is the original point cloud and $\text{sample}()$ initializes $N$ new points denoted as $\boldsymbol{x}_0$. We believe the chosen threats mostly cover the existing attack surfaces on point cloud data. It is also beneficial to study how different threats would affect the adversarial robustness in point cloud recognition under our adversarial training strategies which will be introduced next.

## 2.2 Adversarial Training with Self-Supervisions

We first introduce the chosen 3D self-supervised learning methods, followed by two strategies to incorporate these pretext tasks in adversarial training.

**3D Self-Supervised Learning.** The primary goal of self-supervised learning (SSL) is to learn effective feature representations with unlabeled data. Given a pretext task $P_t$, the pre-training process is still conducted in a supervised manner with self-generated data $\boldsymbol{x}^t$ and label $\boldsymbol{y}^t$ from pristine data $\boldsymbol{x}$, where $(\boldsymbol{x}^t, \boldsymbol{y}^t) = P_t(\boldsymbol{x})$. Therefore, a target loss function $\mathcal{L}_t(\boldsymbol{x}^t, \boldsymbol{y}^t; \mathcal{F}_{\boldsymbol{\theta}_t}^t)$ will be minimized during the optimization, where $\boldsymbol{\theta}_t$ consists of the shared backbone parameters $\boldsymbol{\theta}_m$ and customized branch parameters $\boldsymbol{\theta}_c$ (*i.e.,* $\boldsymbol{\theta}_t := [\boldsymbol{\theta}_m; \boldsymbol{\theta}_c]$). We utilize the following 3D SSL tasks in our study (Figure 1(a)).

• *3D Rotation* [27]: Similar to the rotation task in 2D vision [34], the data and label are generated by rotating the original point clouds to pre-defined angles $\eta$ in the 3D space. Therefore, the problem is to correctly predict 3D rotation angles *w.r.t.* the input point cloud. The objective function can be formulated as a cross-entropy (CE) loss: $\mathcal{L}_{rotation} = \text{CE}(\boldsymbol{x}^{rotation}, \boldsymbol{y}^{rotation}; \mathcal{F}^{rotation})$.

• *3D Jigsaw* [28]: Different from the jigsaw task in 2D vision [35] which is defined as a classification problem, 3D jigsaw solicits a segmentation model. A point cloud is evenly divided to $k^3$ small cubes and shuffled to different positions. Points inside each small cube are assigned to a label signaling its original position. The problem, thus, is to correctly predict the original cube position of each point. Similarly, its objective is to minimize a CE loss: $\mathcal{L}_{jigsaw} = \text{CE}(\boldsymbol{x}^{jigsaw}, \boldsymbol{y}^{jigsaw}; \mathcal{F}^{jigsaw})$.

• *Autoencoder* [29, 36]: An autoencoder utilizes an encoder $z = E(\boldsymbol{x})$ to learn a compact representation and a decoder $D(E(\boldsymbol{x}))$ to reconstruct the point cloud. We utilize different backbones as the encoder $E(\cdot)$ and FoldingNet [29] as the decoder $D(\cdot)$ due to its satisfactory performance. We use three different positional encodings: plane, 3D sphere, and 3D gaussian in our experiments. we use the Chamfer distance [37] as the reconstruction loss: $\mathcal{L}_{ae} = \text{Chamfer}(D(E(\boldsymbol{x}^{ae})), \boldsymbol{x}^{ae}; \mathcal{F}^{ae})$.

The detailed description of the pretext tasks can be found in the supplements.

**Adversarial Pre-training for Fine-tuning (APF).** As introduced in §1, adversarial training (AT) [23, 38, 39] has been demonstrated to be one of the most longstanding and practical defenses. We thus enable AT in both pre-training and fine-tuning stages:

$$\arg\min_{\boldsymbol{\theta}} \quad \mathbb{E}_{(\boldsymbol{x},\boldsymbol{y})\sim\mathcal{D}} \left[ \max_{\sigma\in\mathbb{S}} \mathcal{L}(\boldsymbol{x} + \sigma, \boldsymbol{y}, \boldsymbol{\theta}) \right] \tag{4}$$

where $\mathcal{L} \in \{\mathcal{L}_t, \mathcal{L}_f\}$ for loss functions in pre-training ($t$) and fine-tuning ($f$) stages, $\sigma$ is the adversarial perturbations, and $\mathbb{S}$ represents its manipulation space (*i.e.*, $\ell_\infty$ in our study). AT essentially solves a min-max problem. In the inner loop, the optimizer tries to find adversarial examples that maximize the target loss, and the outer loop updates the network parameters to correctly recognize the generated adversarial examples. In contrast, standard training (ST) is simply to optimize $\arg\min_{\boldsymbol{\theta}} \mathbb{E}_{(\boldsymbol{x},\boldsymbol{y})\sim\mathcal{D}} [\mathcal{L}(\boldsymbol{x}, \boldsymbol{y}, \boldsymbol{\theta})]$.

In the pre-training stage of APF, we leverage both standard and adversarial training to get the pre-trained backbones $\mathcal{M}_{\boldsymbol{\theta}_m}$ and $\mathcal{M}_{\boldsymbol{\theta}_m}^{adv}$. Given a pre-trained backbone parameterized by $\boldsymbol{\theta}_m$, in the second stage, we adversarially fine-tune all $\boldsymbol{\theta}_f := [\boldsymbol{\theta}_m; \boldsymbol{\theta}_h]$ for the recognition task, as illustrated in Figure 1(b). The network branches at the penultimate vector for the rotation task and the first global feature [21] for the jigsaw and autoencoder tasks since they use the segmentation head.

**Adversarial Joint Training (AJT).** Besides pre-training for fine-tuning, joint training is another way to apply SSL. The objective function is formulated as:

$$\arg\min_{\boldsymbol{\theta}_m; \boldsymbol{\theta}_h; \boldsymbol{\theta}_c} \quad \mathbb{E}_{(\boldsymbol{x},\boldsymbol{y})\sim\mathcal{D}} \left[ \max_{\sigma\in\mathbb{S}} \mathcal{L}_f(\boldsymbol{x} + \sigma, \boldsymbol{y}, \boldsymbol{\theta}_f) \right] + \lambda \cdot \mathcal{L}_t(\boldsymbol{x}^t, \boldsymbol{y}^t, \boldsymbol{\theta}_t) \tag{5}$$

where $\lambda$ is a hyperparameter to balance the SSL and recognition tasks. Two tasks share the same backbone $\boldsymbol{\theta}_m$ with two different branches, parameterized by $\boldsymbol{\theta}_h$ and $\boldsymbol{\theta}_c$, respectively. We also enable dual batch normalization [40] in AJT for $\boldsymbol{x}$ and $\boldsymbol{x}^t$ since they should belong to different underlying distributions. We use two model-agnostic tasks, *i.e.,* 3D rotation and jigsaw in AJT.

Similarly, in our AJT analysis, all $\mathcal{L}_t$ and $\mathcal{L}_f$ can be formulated as CE loss. We empirically set $\lambda = 1$ and leverage the same branching point with APF. The whole network is trained to predict the supervised task with the original head and the SSL task with the auxiliary head (Figure 1(b)).

## 3 Experiments and Results

In this section, we present our experimental setups and results. We first introduce the adopted datasets and adversarial settings. We leverage PS as the primary threat in our study since we find it is the most powerful adversary and the other two adversaries easier to detect. Next, we detail our extensive evaluation of two fundamental point cloud recognition tasks: classification and part segmentation.

### 3.1 Evaluation Setups

**Datasets.** We leverage four datasets ($\mathcal{D}$): ModelNet40 [30] (40 classes), ModelNet10 [30] (10 classes), ScanObjectNN [41] (15 classes), and ShapeNetPart [42] throughout our experiments. Specifically, the first three datasets are utilized for the classification task and the last one is for the part segmentation task. ModelNet and ShapeNetPart are captured from CAD models, and ScanObjectNN is scanned and extracted from real-world indoor scenes. For each point cloud, we randomly sample 1024 points and normalize them to an edge-length-2 cube ($[-1, 1]$) for experimentation. We follow the default split of training and test sets in [5] and [43]. For SSL, we randomly sample $\boldsymbol{y}^t$ from the pre-defined label sets and further generate $\boldsymbol{x}^t$ based on $\boldsymbol{y}^t$ in each iteration. Specifically, we choose $\eta = 6, 18$ and $k = 3, 4$ for rotation and jigsaw tasks, followed by the suggestion of [27] and [28].

**Adversary.** As introduced in §2.1, for PS, we exploit 7-step and 200-step $\ell_\infty$ PGD attacks [23] targeting the cross-entropy loss for adversarial training and testing, respectively. We follow Sun *et al.* [21] to empirically set the perturbation boundary $\epsilon = 0.05$ ($||\sigma||_\infty \leq 0.05$) since perturbed

point clouds with $\epsilon = 0.05$ are at the edge of correct human predictions of objects. Numerically, $\epsilon = 0.05$ out of the range [-1,1] is also similar to the commonly used $\epsilon = \frac{8}{255}$ in 2D adversarial training [23]. Note that, different from discrete RGB values in 2D images, point cloud's features (coordinates) are continuous. We thus utilize PGD step size $\alpha = 0.01$ and $\alpha = 0.005$ in the training and testing phases, respectively. We allow a stealthy perturbation of $N = 100$ points for PD and PA since we believe a larger number of modification would be easy to detect [13, 33]. We drop the most $k = 14$ and 5 salient points every iteration in training and testing phases for PD, respectively.

The behind rationale is that a severer attacker is preferred in the testing phase to evaluate the true robust accuracy. The attack setups in PA are the same as those in PS since they utilize similar $\ell_\infty$ PGD perturbation methods. We exploit PS as the primary threat for the classification and part segmentation tasks, and leverage the other two threats to further demonstrate our robustness improvements in the classification task.

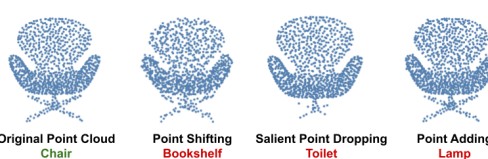

Original Point Cloud — Chair  Point Shifting — Bookshelf  Salient Point Dropping — Toilet  Point Adding — Lamp

Figure 2: Adversarial Examples of Different Threats.

## 3.2 Point Cloud Classification

In this section, we first introduce the setups of our point cloud classification analysis. Next, we introduce the detailed study of APF and AJT. We further evaluate our robust models with unforeseen attacks to demonstrate that our robustness improvements are non-trivial. We find that pre-trained models from different SSL tasks preserve different vulnerabilities; hence we use simple yet powerful ensemble methods to boost the robustness. Lastly, we evaluate our methods in different threats to demonstrate their generality.

**Training Details**. We generally follow data augmentation methods (*e.g.,* jitter and translation) in DGCNN [15] in our study. All pre-trained and fine-tuned models in APF are trained using Adam [44]. We use batch sizes of 32 for PointNet and DGCNN, and 128 for PCT. The initial learning rate is set to 0.001 for PointNet and DGCNN, and $5 \times 10^{-4}$ for PCT. Both pre-training and fine-tuning take 250 epochs, where a $10\times$ decay happens at the 100-th, 150-th, and 200-th epoch. We leverage the same training setups in AJT. All experiments are done on 1 to 4 NVIDIA V100 GPUs [45].

**Model Adaptation to AT.** First, PointNet [5] leverages exponential learning rate decay, and DGCNN and PCT utilize cosine annealing learning rate decay. Through preliminary experiments, we find that a piecewise decay of the learning rate improves the default baseline by a noticeable margin. Second, two T-Nets in PointNet [5] will also make AT unstable since they holistically modify the point cloud and features. We thus remove them in our experiments. Third, PCT by default leverage farthest point sampling (FPS) to sample and group local points, which will cause AT unstable since the perturbed point cloud samples different anchor points in each iteration. We replace FPS with EdgeConv on each point. It is worth noting that such a change will not modify

Table 1: Summarized Results (%) on ModelNet40 by Model Adaptation.

|  | PointNet | | DGCNN | | PCT | |
|---|---|---|---|---|---|---|
|  | CA | RA | CA | RA | CA | RA |
| Default ST | 88.0 | 0.0 | 91.3 | 2.9 | 91.9 | 0.0 |
| Selected ST | 88.7 | 0.3 | 91.5 | 3.2 | 92.1 | 2.3 |
| Default AT | 87.1 | 33.6 | 90.6 | 58.7 | 88.4 | 44.2 |
| Selected AT | 87.7 | 37.9 | 90.6 | 62.0 | 89.7 | 49.1 |

the application of self-attention in PCT. As presented in Table 1[2], the modifications will both improve the standard and adversarial training baseline, and we introduce the details in the supplements.

### 3.2.1 Self-Supervised Pre-training Helps Adversarial Fine-tuning

We systematically evaluate all configurations in APF under PS attack. As introduced earlier, we use standard and adversarial training to get the pre-trained models. From Table 2, we can make several interesting observations. First, we find that our APF strategy generally enhances the adversarial robustness. The best-fine-tuned models achieve 14.2%, 5.4%, and 2.2% robustness improvements in PointNet, DGCNN, and PCT on ModelNet40, respectively. The enhancements on the real-world dataset, ScanObjectNN, *i.e.,* 1.8%, 10.4%, and 6.9% in PointNet, DGCNN, and PCT, are also significant, which demonstrate the generality of APF. Second, we find that DGCNN outstands to be the most robust architecture, consistently achieving $\sim$15% stronger robustness than the other two models on both ModelNet40 and ScanObjectNN. Lastly, jigsaw-based APF offers more robustness improvements than the other two methods while maintaining slightly higher clean accuracy (CA).

---

[2]The 1-st and 2-nd highest accuracy among fine-tuned models in each column are noted, and we use the same mark throughout this paper. CA and RA denote clean and robust accuracy, respectively.

Table 2: Evaluation Results (%) of Adversarial Pre-training for Fine-tuning and Task Ensembles.

| Pretext Task | Parameters | ModelNet40 | | | | | | ScanObjectNN | | | | | | ModelNet10 | | | | | |
|---|---|---|---|---|---|---|---|---|---|---|---|---|---|---|---|---|---|---|---|
| | | PointNet | | DGCNN | | PCT | | PointNet | | DGCNN | | PCT | | PointNet | | DGCNN | | PCT | |
| | | CA | RA | CA | RA | CA | RA | CA | RA | CA | RA | CA | RA | CA | RA | CA | RA | CA | RA |
| AT Baseline | N/A | 87.7 | 37.9 | 90.6 | 62.0 | 89.7 | 49.1 | 69.9 | 23.7 | 74.4 | 30.9 | 72.4 | 20.5 | 96.6 | 79.7 | 98.1 | 86.3 | 97.4 | 80.0 |
| 3D Rotation | $\eta = 6$ | 87.2 | 48.0 | 91.4 | 63.6 | 90.2 | 50.7 | 69.1 | 24.5 | 75.7 | 32.9 | 72.6 | 20.6 | 96.8 | 79.0 | 97.7 | 84.9 | 97.2 | 80.4 |
| | $\eta = 18$ | 87.2 | 48.3 | 91.1 | 64.1 | 90.2 | 49.5 | 69.5 | 25.0 | 73.8 | 32.2 | 72.5 | 20.1 | 97.1 | 79.3 | 98.5 | 85.3 | 97.8 | 80.3 |
| Adversarial 3D Rotation | $\eta = 6$ | 87.6 | 42.1 | 90.8 | 61.8 | 90.4 | 50.8 | 69.6 | 25.3 | 75.0 | 36.8 | 71.6 | 28.7 | 97.0 | 79.9 | 97.7 | 87.5 | 98.0 | 82.2 |
| | $\eta = 18$ | 87.4 | 45.7 | 90.9 | 62.9 | 90.4 | 50.1 | 69.3 | 24.5 | 75.0 | 36.3 | 73.1 | 26.9 | 97.0 | 79.7 | 98.0 | 88.2 | 97.4 | 83.7 |
| 3D Jigsaw | $k = 3$ | 87.6 | 50.1 | 90.0 | 67.4 | 90.4 | 51.1 | 70.8 | 25.5 | 79.0 | 33.8 | 73.4 | 23.2 | 96.8 | 80.0 | 98.0 | 89.6 | 97.8 | 81.5 |
| | $k = 4$ | 87.6 | 50.9 | 90.1 | 65.3 | 90.3 | 50.2 | 70.2 | 25.4 | 76.2 | 35.3 | 73.8 | 24.6 | 96.7 | 80.2 | 98.0 | 89.0 | 97.7 | 81.9 |
| Adversarial 3D Jigsaw | $k = 3$ | 88.2 | 52.1 | 89.6 | 65.8 | 89.8 | 51.3 | 69.0 | 24.8 | 77.5 | 41.3 | 72.5 | 26.3 | 97.0 | 80.6 | 98.5 | 90.5 | 97.4 | 83.5 |
| | $k = 4$ | 87.8 | 50.5 | 89.9 | 65.3 | 89.6 | 51.0 | 69.9 | 25.5 | 76.1 | 40.6 | 73.1 | 27.4 | 97.0 | 80.5 | 98.0 | 89.1 | 97.3 | 83.9 |
| Autoencoder | sphere | 87.4 | 50.0 | 89.9 | 62.8 | 90.2 | 50.7 | 69.9 | 25.1 | 76.1 | 36.0 | 71.3 | 24.1 | 97.0 | 80.5 | 98.2 | 86.8 | 97.1 | 80.1 |
| | plane | 87.1 | 48.8 | 90.1 | 62.2 | 90.2 | 50.2 | 69.4 | 25.5 | 76.2 | 35.6 | 71.1 | 22.6 | 96.8 | 80.8 | 97.8 | 87.6 | 97.0 | 80.1 |
| | gaussian | 87.4 | 48.9 | 90.8 | 63.3 | 89.7 | 50.3 | 69.7 | 23.8 | 75.6 | 35.8 | 71.3 | 24.8 | 96.8 | 80.5 | 97.8 | 86.4 | 97.1 | 80.1 |
| Adversarial Autoencoder | sphere | 87.1 | 49.7 | 90.0 | 62.2 | 90.3 | 50.0 | 70.4 | 25.2 | 75.2 | 36.2 | 72.6 | 22.2 | 96.7 | 80.4 | 97.5 | 87.3 | 97.5 | 82.1 |
| | plane | 86.9 | 46.6 | 89.7 | 61.8 | 89.7 | 50.0 | 69.2 | 24.0 | 75.6 | 38.0 | 73.3 | 21.6 | 97.0 | 80.6 | 98.0 | 86.1 | 97.7 | 82.5 |
| | gaussian | 87.1 | 48.5 | 90.7 | 62.7 | 90.2 | 50.5 | 68.8 | 25.0 | 74.7 | 36.3 | 72.6 | 23.4 | 97.0 | 80.2 | 97.8 | 88.4 | 97.4 | 83.2 |
| Max Ensemble | | 88.5 | 53.5 | 91.4 | 69.4 | 90.5 | 55.4 | 70.8 | 27.3 | 79.1 | 42.6 | 74.0 | 28.9 | 97.2 | 82.5 | 98.5 | 91.0 | 98.1 | 85.2 |
| Mean Ensemble | | 88.4 | 52.5 | 91.4 | 68.7 | 90.4 | 57.9 | 70.8 | 26.9 | 79.0 | 41.9 | 74.2 | 28.4 | 97.0 | 82.4 | 98.6 | 90.9 | 98.0 | 84.9 |

Specifically, jigsaw-based APF, on average, further boosts DGCNN's robust accuracy (RA) by 2.8%, 2.2%, and 2.7% on three datasets, respectively.

**Insights**. Different from 2D images that possess both texture and shape information, 3D point clouds naturally bias towards shape. In 2D image space, it is widely recognized that local and global features correspond to the texture and shape information, respectively [46]. Recent studies have demonstrated that appreciation of global/shape features can help improve model robustness on image classification [47]. However, we find some distinctions in point cloud recognition. As mentioned above, PointNet with only global feature learning will be easily affected by the perturbed points (Table 2). Due to the sparsity of point clouds, the local feature actually represents the smoothness of the object's surface. Thus, learning robust local features is critical for correctly recognizing a perturbed point cloud, as it limits the adversarial effect propagation to the model output. We also include a preliminary study on contrastive pre-training [48] and find that it can be viewed as a global feature learning scheme as well. Detailed results can be found in the supplements.

As summarized above, DGCNN achieves the strongest robustness under AT, attributed to the hierarchical usage of EdgeConv [15]. EdgeConv dynamically aggregates local features by exploiting $k$NN. Such an aggregation method has the ability to calibrate the adversarial effect in the local feature learning stage. Although transformer-based architectures have gained tremendous visibility recently [49], we find that PCT does not have a major robustness improvement compared to PointNet. Self-attention increases the capacity of the model architecture, but it also enlarges the receptive field of the model [50]. In PCT, each point can influence every other point's feature, which will potentially increase the model's fragility [51].

Moreover, we also find that jigsaw-based APF is the most effective method to improve adversarial robustness, aligning well with our above insights. Jigsaw SSL makes the model learn to reassemble the randomly displaced local point clusters, where the model is enforced to learn the displaced local features. Meanwhile, to correctly reconstruct the point cloud, jigsaw SSL also requires the model to capture the global and holistic semantics. Nevertheless, rotation and autoencoder-based pre-training methods focus more on global feature learning. Therefore, we believe jigsaw-based APF is a perfect candidate to strengthen the association between local and global features in point cloud learning, hence improving the adversarial robustness under APF.

### 3.2.2 Adversarial Joint Training Does not Always Improve Robustness

We further analyze the implication of SSL tasks in AJT. As presented in Table 3, AJT can still enhance the robustness in PointNet and DGCNN. For instance, AJT improves their RA by 1.8% and 8.0% on ScanObjectNN, respectively. However, AJT overall cannot outperform APF in point cloud recognition. Especially, we find AJT even degrades the RA of PCT compared to the standard AT.

**Insights**. We find this also to be related to the natural characteristic of point cloud data. Although SSL can help models learn strong priors and context information, it is still a *separate* learning task. Rotated and disassembled images still preserve similar local features to the original images since the

Table 3: Evaluation Results (%) of Adversarial Joint Training.

| Pretext Task | Parameters | ModelNet40 | | | | | | ScanObjectNN | | | | | | ModelNet10 | | | | | |
|---|---|---|---|---|---|---|---|---|---|---|---|---|---|---|---|---|---|---|---|
| | | PointNet | | DGCNN | | PCT | | PointNet | | DGCNN | | PCT | | PointNet | | DGCNN | | PCT | |
| | | CA | RA | CA | RA | CA | RA | CA | RA | CA | RA | CA | RA | CA | RA | CA | RA | CA | RA |
| AT Baseline | N/A | 87.7 | 37.9 | 90.6 | 62.0 | 89.7 | 49.1 | 69.9 | 23.7 | 74.4 | 30.9 | 72.4 | 20.5 | 96.6 | 79.7 | 98.1 | 86.3 | 97.4 | 80.0 |
| 3D Rotation | $\eta=6$ | 86.8 | 45.0 | 91.2 | 60.7 | 89.5 | 44.3 | 67.8 | 24.3 | 74.2 | 37.8 | 72.3 | 20.3 | 96.6 | 79.0 | 98.1 | 86.3 | 97.8 | 73.8 |
| | $\eta=18$ | 86.5 | 46.4 | 91.3 | 62.0 | 88.9 | 42.9 | 68.7 | 25.1 | 76.2 | 37.2 | 72.1 | 19.8 | 97.0 | 79.9 | 97.9 | 85.7 | 98.1 | 75.6 |
| 3D Jigsaw | $k=3$ | 87.6 | 42.5 | 91.0 | 62.3 | 90.2 | 43.1 | 69.4 | 25.5 | 77.1 | 38.9 | 72.1 | 20.7 | 96.8 | 79.8 | 98.4 | 87.9 | 97.7 | 76.8 |
| | $k=4$ | 87.2 | 46.7 | 91.1 | 61.7 | 89.8 | 40.9 | 70.0 | 24.6 | 75.9 | 38.4 | 73.7 | 20.8 | 96.8 | 77.9 | 98.0 | 88.6 | 97.1 | 78.0 |
| Autoencoder | sphere | 87.5 | 44.4 | 90.9 | 62.1 | 89.6 | 49.2 | 68.9 | 24.2 | 75.5 | 36.5 | 72.5 | 20.5 | 96.7 | 79.8 | 98.2 | 86.3 | 97.5 | 80.3 |
| | plane | 87.4 | 42.1 | 90.7 | 61.9 | 89.3 | 48.7 | 68.5 | 23.9 | 75.6 | 34.7 | 72.8 | 20.6 | 96.7 | 79.7 | 98.1 | 86.2 | 97.4 | 79.9 |
| | gaussian | 86.9 | 43.9 | 90.9 | 61.9 | 88.9 | 49.2 | 68.7 | 24.4 | 76.3 | 35.1 | 72.1 | 20.5 | 96.6 | 79.7 | 98.3 | 86.9 | 97.5 | 80.0 |

RGB values do not change, so that the auxiliary optimization in AJT will not distract AT but help models learn robust global features [24]. However, point cloud models take point coordinates $xyz$ as input. Rotated and disassembled point clouds have significant variations in their coordinates' numeric values. Although we apply dual batch normalization [52] to migrate the feature heterogeneous problems, such discrimination will consequently distract model learning in AJT, and thus hurt the RA performance. The usage of self-attention in PCT will further expand this impact since it introduces a global receptive field [51]. We find that the results of AJT using autoencoders are more stable than the other two tasks. We believe it is because the input for the autoencoder is the same as the recognition task so that the distributional gap is small. However, it is still worse than the pre-training scheme.

### 3.2.3 Robustness against Unforeseen Attacks and Noises

We have so far shown that APF generally improves the robustness of point cloud classification. In this section, we leverage unforeseen attacks to further demonstrate that the enhancements are non-trivial. We select the fine-tuned models with the highest RA in Table 2, and all results here are averaged from five runs using different random seeds.

We leverage unforeseen attacks: Auto Attack (AA) [53], Momentum Iterative Method (MIM) [54], and PGD with both cross-entropy and margin loss (*i.e.,* $\mathcal{L}_{\text{margin}} = Z(\boldsymbol{x})_y - \max_{i \neq y} Z(\boldsymbol{x})_i$, where $Z(\boldsymbol{x})$ is the logit output of the target classifier). We elaborate the details of AA (*i.e.,* A-PGD) and MIM in the supplements. All attacks use 200 steps to find the potential adversarial examples. As shown in Figure 3(a) and 3(b), the best fine-tuned models consistently achieve higher RA than models from standard AT with 2.6% - 15.3% and 1.0% - 10.1% improvements on ModelNet40 and ScanObjectNN, respectively. We find that DGCNN still outperforms the other two models achieving the highest RA even against the strongest Auto Attacks [53].

Moreover, we exploit the transfer attack to confirm the effectiveness of APF. We fine-tune five models using the same (best) setting but different random seeds for transfer-based attacks and test the transferability of generated adversarial examples among different models. As Figure 3(d) and 3(e) show, adversarial examples cannot transfer well and there is a ~10% gap in RA compared to attacks on target models, which further validates the real robustness improvements.

We also find that Gaussian and uniform noises are ineffective in breaking model robustness, and the RA is therefore close to the CA. As shown in Figure 3(d) and 3(e), the fine-tuned models still outperform standard AT baseline by 0.1% - 2.2% and 1.0% - 5.7% on ModelNet40 and ScanObjectNN, respectively, in RA. More evaluation results of our adversarial training strategy are elaborated in the supplements.

### 3.2.4 Attack Transferability and Task Ensemble

Since we leverage three pretext tasks for self-supervised pretraining, we also test the transferability of adversarial examples generated by models pretrained on different SSL tasks. As shown in Figure 4, different fine-tuned models preserve different vulnerabilities, and there is at least ~10% gain of RA when transferring attacks on mod-

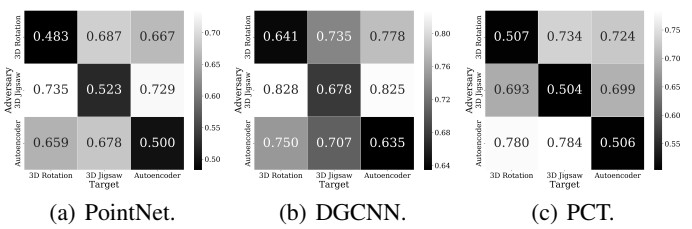

(a) PointNet.  (b) DGCNN.  (c) PCT.

Figure 4: Robust Accuracy on Transfer Attacks among Fine-tuned Models from Different SSL Tasks on ModelNet40.

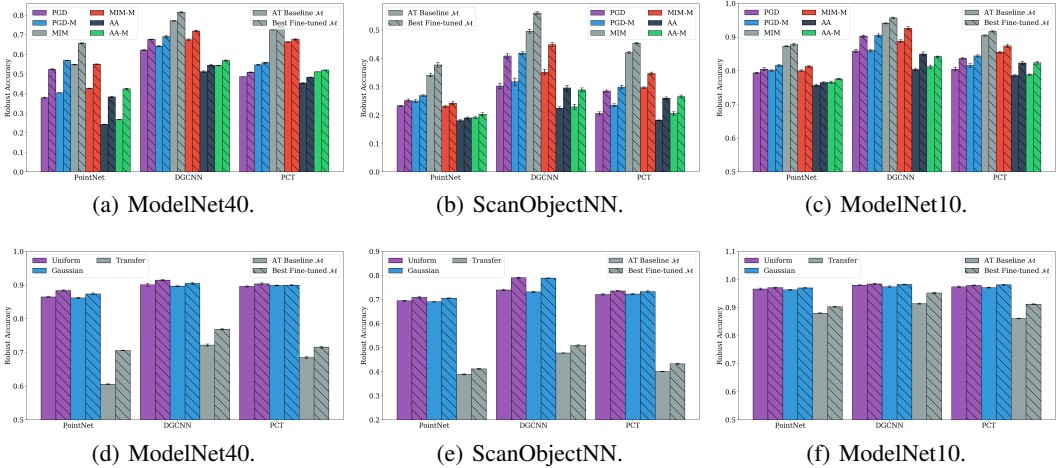

Figure 3: Evaluation Results of Unforeseen Attacks. Plots in the first row present the evaluation results of unforeseen adversarial attacks, where "-M" denotes the attacks on margin loss. Plots in the second row present the evaluation results of noise and transfer attacks.

els fine-tuned from different tasks on ModelNet40. We thus combine the models and further test the ensemble model's robustness. We use two ensemble methods, which are taking the max and mean value of the stacked logits from individual models *i.e.,* $L^i_{ensemble} = \max/\text{mean}([L^i_{rotation}, L^i_{jigsaw}, L^i_{autoencoder}])$ where $L^i$ denotes the $i$-th value in the logit L. By doing so, we form a "wider" model where the gradients can still propagate smoothly back to the input. Therefore, we also exploit the 200-step PGD attack to test their robustness. As presented in the last two rows of Table 2, both ensemble models can further boost the RA by a significant margin while maintaining similar clean accuracy. For example, our best ensemble models achieve robust accuracy of 53.5% (+15.6%), 69.4% (+7.4%) and 57.9% (+8.8%) with PointNet, DGCNN, and PCT on ModelNet40 respectively. It is worth noting that our fine-tuning strategy does not enforce a diversity regulation, unlike [25], which further demonstrates that pre-training on different pretext tasks indeed brings distinct context information into pre-trained models.

### 3.2.5 Robustness against Other Point Cloud Threats

As mentioned before, we also consider two other threats that specifically target point cloud recognition. As we have demonstrated that jigsaw-based APF reaches the best robustness enhancement under point shifting (PS), we replace PS attack in AT with salient point dropping (PD) and point adding (PA) to test the effectiveness of jigsaw-based APF strategy.

Table 4: Evaluation (%) of PD on ScanObjectNN.

| Pretext Task | Parameters | PointNet | | DGCNN | | PCT | |
|---|---|---|---|---|---|---|---|
| | | CA | RA | CA | RA | CA | RA |
| ST Baseline | N/A | 69.8 | 48.5 | 79.3 | 60.9 | 74.2 | 56.1 |
| AT Baseline | N/A | 70.9 | 64.2 | 76.2 | 77.1 | 73.3 | 73.7 |
| 3D Jigsaw | $k = 3$ | 71.3 | 65.2 | 81.4 | 80.0 | 76.8 | 76.9 |
| | $k = 4$ | 72.5 | 66.3 | 81.2 | 80.7 | 77.8 | 80.1 |
| Adversarial | $k = 3$ | 71.1 | 64.7 | 78.7 | 79.5 | 75.4 | 76.2 |
| 3D Jigsaw | $k = 4$ | 70.9 | 65.7 | 79.3 | 80.4 | 75.9 | 78.3 |

Table 5: Evaluation (%) of PA on ScanObjectNN.

| Pretext Task | Parameters | PointNet | | DGCNN | | PCT | |
|---|---|---|---|---|---|---|---|
| | | CA | RA | CA | RA | CA | RA |
| ST Baseline | N/A | 69.8 | 57.6 | 79.3 | 38.0 | 74.2 | 39.6 |
| AT Baseline | N/A | 69.0 | 59.4 | 77.4 | 63.5 | 72.1 | 56.8 |
| 3D Jigsaw | $k = 3$ | 70.2 | 59.9 | 82.3 | 66.3 | 75.7 | 60.4 |
| | $k = 4$ | 70.4 | 60.6 | 83.1 | 66.6 | 76.4 | 58.7 |
| Adversarial | $k = 3$ | 70.4 | 62.5 | 77.5 | 67.8 | 75.2 | 64.4 |
| 3D Jigsaw | $k = 4$ | 70.7 | 59.4 | 79.5 | 68.2 | 74.4 | 65.5 |

We find that jigsaw-based APF can still strengthen the robustness under these two threats on three datasets. Especially, as shown in Table 4 and 5, DGCNN still achieves the highest RA, which on average outperforms PointNet and PCT by 14.8% and 2.3% in PD-AT, and 6.6% and 4.9% in PA-AT, respectively, which is consistent with our previous findings. Moreover, as PD and PA tend to be much weaker adversaries than the PS attack, the enhancements mainly appear in the challenging real-world dataset, ScanObjectNN, where jigsaw-based APF boosts the RA of DGCNN by 3.6% and 4.7% under PD and PA. The key insights still hold in ModelNet datasets under these two threats, and we present the detailed results in the supplements due to space constraints.

## 3.3 Point Cloud Part Segmentation

In addition to object classification, we conduct the *first* study on analyzing the robustness of the point cloud part segmentation on ShapeNetPart [42] with 17 classes of 3D objects. Apart from semantic segmentation that all samples from a dataset share the same set of labels, part segmentation task assigns a unique label set for each class of objects. For instance, the class of chair has the label set: {seat, back, arm, leg} and the airplane has {wing, body, tail, engine}, as illustrated in Figure 5. We use the same backbone with a segmentation head for this task and mean intersection-over-union (mIoU) [43] as the evaluation metric. Especially since each class possesses its own label set, the models also take the category (one-hot vector) as input. As we find APF is an effective and efficient strategy, we use the PS attack with the same AT and APF setups as the classification task to train the models (§ 3.1).

Table 6: Evaluation Results (%) of the Part Segmentation Task.

| Pretext Task | Parameters | PointNet | | DGCNN | | PCT | |
|---|---|---|---|---|---|---|---|
| | | C-mIoU | R-mIoU | C-mIoU | R-mIoU | C-mIoU | R-mIoU |
| ST Baseline | N/A | 83.2 | 32.5 | 84.2 | 42.0 | 84.2 | 34.6 |
| AT Baseline | N/A | 79.2 | 62.7 | 82.9 | 69.5 | 82.6 | 67.8 |
| 3D Rotation | $\eta = 6$ | 79.1 | 63.0 | 82.8 | 69.3 | 82.8 | 67.7 |
| | $\eta = 18$ | 79.3 | 63.0 | 82.9 | 69.5 | 82.9 | 67.5 |
| Adversarial 3D Rotation | $\eta = 6$ | 79.3 | 62.9 | 82.9 | 69.2 | 82.2 | 67.2 |
| | $\eta = 18$ | 79.1 | 62.6 | 82.9 | 69.0 | 81.0 | 65.7 |
| 3D Jigsaw | $k = 3$ | 79.4 | 63.0 | 82.7 | 70.1 | 82.8 | 68.1 |
| | $k = 4$ | 79.4 | 63.1 | 82.9 | 70.3 | 82.4 | 68.4 |
| Adversarial 3D Jigsaw | $k = 3$ | 79.7 | 62.5 | 82.8 | 69.7 | 81.1 | 65.8 |
| | $k = 4$ | 79.9 | 62.7 | 82.7 | 70.3 | 82.0 | 66.7 |

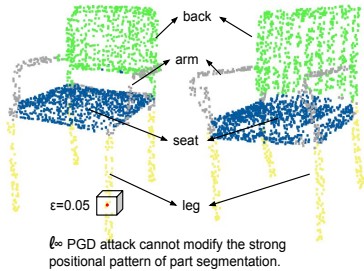

Figure 5: Illustration of the Point Cloud Part Segmentation Using "Chair".

Table 6[3] summarizes the quantitative results. To our surprise, we find that part segmentation is a more robust task than classification. The robust mIoU is relatively high even in the ST baseline. Moreover, DGCNN still achieves the highest clean and robust mIoU in part segmentation, further confirming its supremacy in point cloud learning. Furthermore, jigsaw-based APF can also promote the robustness of part segmentation by 0.4%, 0.8%, and 0.6% in PointNet, DGCNN, and PCT, respectively.

**Insights**. We attribute the natural robustness of part segmentation to the intrinsic feature of point clouds. Since all point clouds in ShapeNetPart are sampled from CAD models, they are posed to face the same direction. Thus, models may memorize the 3D space corresponding to the specific label during learning. Taking the class of chair as an example, the points whose $z \in [-1, 0]$ most likely belong to the "leg" label. However, $\ell_\infty$-based adversary with $\epsilon = 0.05$ might not be able to perturb this strong pattern (Figure 5). We also envision that the robustness of part segmentation may help and transfer to other complex learning tasks if appropriately used. Moreover, DGCNN with jigsaw-based APF still delivers the best performance, which further generalizes our previous findings on the importance of balancing local and global features in point cloud learning.

## 4 Related Work

In this section, we review a few topics related to the adversarial robustness of 3D perception: 3D deep learning, self-supervised learning, and adversarial attacks and defenses.

### 4.1 Deep Learning on 3D Point Clouds

In image-based perception, there has been stellar progress on architectures and learning algorithms for convolutional neural networks. However, in 3D perception, there has been no consensus on the type of models for 3D data partially because there is no standard data format for 3D perception. Yet, point clouds are the most commonly used data format because raw data from both 3D scanners and triangular meshes can easily be converted into a point cloud. Some of the earlier 3D networks use dense voxel grids for perception [1, 2, 55, 56], which discretize a point cloud to voxel grids for classification, segmentation, and object detection. Later, to overcome cubic memory complexity, various types of data structures have been proposed: octrees [3, 4], surfaces [57], graphs [15, 58], point sets [5, 14, 17], and sparse tensors [6, 16]. In this work, we focus on 3D point cloud perception with PointNet [5], DGCNN [15], and PCT [17] as our evaluation backbones since they are widely used and achieve state-of-the-art results for point cloud recognition [59].

---

[3]C-mIoU and R-mIoU denote clean and robust mean intersection-over-union, respectively.

## 4.2 Self-Supervised Learning Approaches

Self-supervised or unsupervised learning (SSL) techniques have shown effective in learning powerful representation without manual efforts, where the labels are generated from the data itself [60–62]. The embeddings pre-trained through self-supervision could be utilized for fine-tuning multiple downstream supervised tasks with better generalization and calibration [63, 64]. Because of the impressive success of SSL in 2D computer vision [34], recent studies have also proposed several self-supervisions for 3D point clouds including autoencoders [29], generative adversarial networks (GANs) [36], 3D rotation [27], and 3D jigsaw [28]. Those primitives hold promises to improve the clean accuracy of point cloud learning. This work systematically explores whether and how SSL could be leveraged to improve the adversarial robustness in point cloud recognition.

## 4.3 Adversarial Attacks and Defenses

Despite the accomplishments that DNNs have achieved, adversarial attacks [65] are becoming the major obstacle in real-world deep learning deployments, especially in safety-critical areas [11, 12, 66–69]. Numerous attacks have been widely studied for various tasks in the 2D [70–76] and 3D [13, 77–79] domains. To address this problem, many defense methods have been proposed to enhance the robustness against adversarial attacks in the 2D [80–88] and 3D domains [19–21]. However, most of them including adding randomization [20, 89, 90], model distillation [83], adversarial detection [84], and input transformation [19, 80–82, 91] have been compromised by adaptive attacks [21, 22, 92]. Certified methods are recently applied to 3D, but they focus on threat models with modified number of points [93] and isometric transformations [94], which are inapplicable to $\ell_\infty$ norm. Adversarial training (AT) [23, 38, 39, 95], on the other hand, provides one of the most longstanding and practical defenses. Various methods have been proposed to improve AT in the 2D domain [52, 96, 97]. Jeddi *et al.* establish a fast adversarial fine-tuning scheme [98]. Hendrycks *et al.* [24] have shown to jointly train the adversarial loss and SSL loss to improve the robustness. Chen *et al.* [25] have applied adversarial pre-training and fine-tuning strategy to improve the robustness. In this work, we explore to apply and improve AT in 3D point cloud recognition.

## 5 Conclusion

In this work, we systematically explore the impact of self-supervised learning (SSL) on the adversarial robustness in 3D point cloud recognition. We find tangible robustness improvements by the adversarial pre-training for fine-tuning strategy. We also experimentally show that robust local features are critical to achieving robustness in 3D, explaining the success of DGCNN and the jigsaw proxy task. Our results shed light for future research on designing more robust models and SSL schemes for 3D point clouds. By providing empirical evaluations on the robustness, our study also motivates future studies in developing theoretical guarantees on the robustness in the 3D domain.

## 6 Acknowledgements

We appreciate our area chairs and anonymous reviewers for their insightful comments. We thank Qingzhao Zhang for proofreading our manuscript. Jiachen Sun thanks Zhao Su for her considerate care and help during COVID-19. This project is partially supported by NSF grants CMMI-2038215 and CNS-1930041.

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
