# Adversarially Robust 3D Point Cloud Recognition Using Self-Supervisions

## Supplementary Materials

**Jiachen Sun** [*1], **Yulong Cao** [1], **Christopher Choy** [2], **Zhiding Yu** [2], **Anima Anandkumar** [2,3], **Z. Morley Mao** [1], **and Chaowei Xiao** [2,4]

[1]University of Michigan , [2]NVIDIA , [3]Caltech , [4]ASU

## A   Implementation Detail

In this section, we introduce our implementation details of the adopted model architectures and self-supervised learning tasks.

### A.1   Model Architecture

As we summarized in § 1, 3D point cloud recognition models could be mainly categorized into three groups, including shared multi-layer-perceptron (MLP)-based, convolution-based, and transformer-based architectures. We select three representatives: PointNet [1], DGCNN [2], and PCT [3] in our study. We utilize the open-sourced codebase in [4], [5], and [6] as our base implementations and follow their MIT licenses to use the codes.

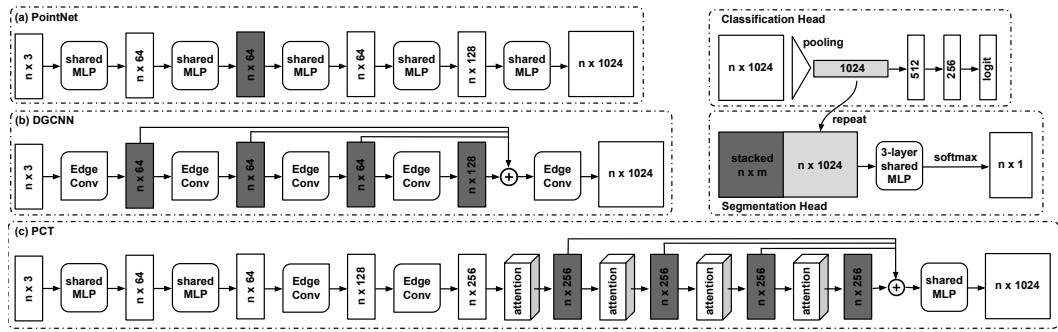

Figure A: Illustration of Different Backbone Architectures.

**PointNet**.  We adopt the same architecture as the original implementation except for the T-Nets, because the isometric robustness is out of the scope of this paper, and T-Nets will make the adversarial training unstable. PointNet leverages a shared MLP to extract features:

$$\boldsymbol{x}_i' = \mathrm{ReLU}(\boldsymbol{\theta}_p \cdot \boldsymbol{x}_i) \tag{S1}$$

The detailed parameters are illustrated in Figure A.

**DGCNN**.  We also adopt the same architecture as the original implementation, as illustrated in Figure A. DGCNN leverages EdgeConv as their basic operation to extract features. EdgeConv finds

---

[*]Correspondence to `jiachens@umich.edu`

35th Conference on Neural Information Processing Systems (NeurIPS 2021)

the $k = 20$ neighbor points and uses the symmetric function to aggregate the local features:

$$\boldsymbol{x}_i' = \max_{j \in \boldsymbol{\zeta}_i^k} [\text{ReLU}(\boldsymbol{\theta}_q \cdot (\boldsymbol{x}_j - \boldsymbol{x}_i) + \boldsymbol{\theta}_p \cdot \boldsymbol{x}_i)] \tag{S2}$$

where $\boldsymbol{\zeta}_i^k$ represents the neighbor points of $\boldsymbol{x}_i$ and the max value is extracted from each feature channel to aggregate the local feature. The EdgeConv layers are stacked to form the DGCNN backbone.

**PCT**. As introduced in § 3.2, we replace the PointNet++-based operation [7] with EdgeConv [2] in the encoding phase of PCT because PointNet++ uses Farthest Point Sampling (FPS). In FPS, different anchor points are sampled from each iteration, so that the local features will not be consistent in adversarial training. EdgeConv generalizes the PointNet++ operation, which aggregates local features from each point without sampling [2]. We adopt EdgeConv with $k = 32$ in PCT. The detailed parameters are illustrated in Figure A.

Please refer to our codebase for detailed parameters like batch normalization and activation functions.

### A.2 Self-Supervised Learning Task

We follow exactly the same setting as Poursaeed *et al.* [8] and Sauder *et al.* [9] for 3D rotation and jigsaw proxy tasks since they are model-agnostic.

**3D Rotation**. For rotation angles $\eta = 6$, we use directions as $\pm x$, $\pm y$, and $\pm z$ axes. For $\eta = 18$, we evenly choose six directions in $xy$, $xz$, and $yz$ planes, as shown in Figure B.

**3D Jigsaw**. As introduced in § 2.2, we choose $k = 3, 4$ in this task. Therefore, the edge-length-2 3D space will be into $3^3 = 27$ and $4^3 = 64$ small cells $\mathcal{C} = \{c_i\}_{i=0}^{k^3-1}$. Each point in cell $c_i$ will be assigned a label as $i$. We randomly permute the cells in the 3D space and make the model predict the segmentation label assigned to each point.

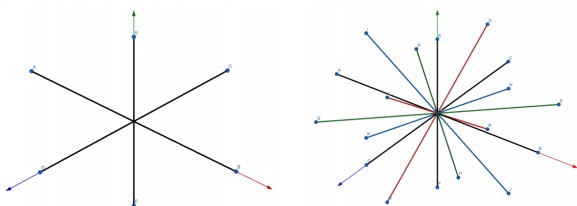

(a) Pre-defined Angle $\eta = 6$.  (b) Pre-defined Angle $\eta = 18$.

Figure B: Illustration of 3D Rotation Pre-defined Angles [8].

**Autoencoder**. For the autoencoder task, as mentioned in § 2.2, we leverage the FoldingNet [10] decoder in our study. We use PointNet [1], DGCNN [2], and PCT [3] backbones as the encoder, respectively. We illustrate the FoldingNet architecture in Figure C. As it shows, there is a positional encoding in the decoding phase, and we leverage 2D plane ($m \times 2$), 3D Gaussian ($m \times 3$), and 3D sphere ($m \times 3$) [11]. Positional encoding serves as a prior to help achieve better reconstructions.

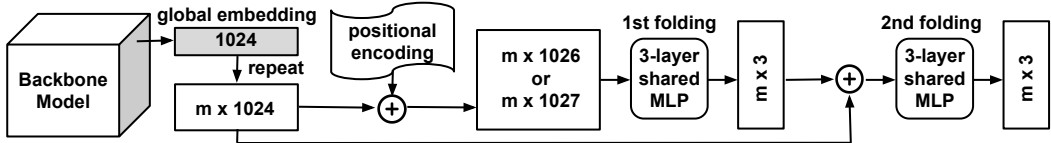

Figure C: Illustration of FoldingNet Architecture.

We leverage Chamfer distance [12] to bound the reconstruction loss:

$$\text{Chamfer}(\boldsymbol{x}, \boldsymbol{x}') = \frac{1}{||\boldsymbol{x}'||_0} \sum_{y \in \boldsymbol{x}'} \min_{x \in \boldsymbol{x}} ||x - y||_2^2 \tag{S3}$$

Although Chamfer distance does not hold for the triangle inequality, it can be still used as the objective for the autoencoder task, and we use the following formulation in the adversarial pre-training of the autoencoder task:

$$\arg\min_{\boldsymbol{\theta}_E} [\max_{\sigma} ||E(\boldsymbol{x}) - E(\boldsymbol{x} + \sigma)||_1] + \arg\min_{[\boldsymbol{\theta}_E; \boldsymbol{\theta}_D]} \text{Chamfer}(\boldsymbol{x}, D(E(\boldsymbol{x}))) \tag{S4}$$

where $E(\cdot)$ and $D(\cdot)$ represent the encoder and decoder, respectively, and $\boldsymbol{\theta}_E := \boldsymbol{\theta}_m$. By doing so, we shift the adversarial training focus from the whole encoder-decoder architecture to the encoder/backbone only, since our goal is to enhance the backbone's robustness.

# B  Evaluation Detail

In this section, we first introduce the detailed formulations of the adopted attack methods. Then, we present additional evaluation results.

## B.1  Attack Method

We introduce the detailed formulation of attack methods used in our study.

**Salient Point Dropping**. We follow the attack setups in [13] to formulate our attack. We first calculate the median "point" of the point cloud $x_{median\,j} = \text{median}(\{x_{ij}|x_i \in \boldsymbol{x}\})(j \in \{1,2,3\})$ as an anchor. Next, the gradients of the coordinates can be calculated as:

$$\frac{\partial \mathcal{L}}{\partial r_i} = \sum_{j=1}^{3} \frac{\partial \mathcal{L}}{\partial x_{ij}} \cdot \frac{x_{ij} - x_{median\,j}}{r_i} \tag{S5}$$

where $r_i = \sqrt{\sum_{j=1}^{3}(x_{ij} - x_{median\,j})^2}$ is the radius of each point in the sphere coordinate. The saliency scores are further calculated by:

$$\text{saliency}(x_i) = -\frac{\partial \mathcal{L}}{\partial r_i} \cdot r_i^{1+\alpha} \tag{S6}$$

where $\alpha = 1$ is chosen, suggested by [13]. As introduced in § 2.1, we drop $k = 14, 5$ points with highest saliency scores in each iteration of the training and testing phases, respectively, until a total drop of $N = 100$ points.

**Momentum Iterative Method**. The Momentum Iterative Method (MIM) attack introduces a momentum term into the adversarial optimization:

$$\boldsymbol{g}_{s+1} = \mu \cdot \boldsymbol{g}_s + \frac{\nabla_{\boldsymbol{x}_s}\mathcal{L}(\boldsymbol{x}_s, \boldsymbol{y}; \mathcal{F})}{||\nabla_{\boldsymbol{x}_s}\mathcal{L}(\boldsymbol{x}_s, \boldsymbol{y}; \mathcal{F})||_1}$$
$$\boldsymbol{x}_{s+1} = \boldsymbol{x}_s + \alpha \cdot \text{sign}(\boldsymbol{g}_{s+1}) \tag{S7}$$

We empirically set $\mu = 1$, followed by [14]. The other setups are similar to PGD attacks.

**Auto Attack**. We adopt the A-PGD attack in Auto Attack (AA) [15]. We follow exactly the same implementation in [16], which is the official codebase of AA.

## B.2  Additional Experiments and Results

In this section, we introduce the evaluation results which are not presented in the main paper due to space constraints.

### B.2.1  Model Adaptation to Adversarial Training

As introduced in § 3.2, we leverage some tricks to improve the standard adversarial training baseline [17]. We provide insights on how different components contribute to the overall improvements. First, the default T-Net [1] in PointNet can be regarded as a self-attention module with restricted capacity since it learns a transformation matrix $\boldsymbol{T}_{n \times n}$ to holistically transform the feature: $\boldsymbol{F}'_{n \times m} = \boldsymbol{F}_{n \times m} \times \boldsymbol{T}_{m \times m}$, where $\boldsymbol{F}'_{n \times m}$ and $\boldsymbol{F}_{n \times m}$ denote the output and input features. The

Table A: PointNet Adaptation Results (%) on ModelNet40.

| Model | CA | RA |
|---|---|---|
| PointNet | $87.2 \pm 0.27$ | $34.1 \pm 5.59$ |
| - T-Net | $87.6 \pm 0.15$ | $37.9 \pm 3.76$ |

adversarial perturbations on $\boldsymbol{T}_{m \times m}$, therefore, will have a significant impact on the downstream features, which will make adversarial training unstable. The primary goal of T-Net is to improve the isometric stability of PointNet. Since isometry is out of the scope of our study, we do not utilize T-Net in our PointNet implementation. Second, the feature extraction in PCT leverages PointNet++ [7], which uses sampling and grouping for local feature aggregation. As introduced in § 3.2 and § A.1, Farthest Point Sampling (FPS) is leveraged to sample the anchor points. However, since the point cloud is perturbed every iteration, different anchor points will be sampled in the learning procedure, making the adversarial training ineffective. As demonstrated in [2], EdgeConv generalizes PointNet++ operation with a larger capacity. EdgeConv can be at high-level viewed as a PointNet++ operation that samples every point in the point cloud. We thus adopt two layers of EdgeConv in the PCT implementation.

### B.2.2 Preliminary Study on Contrastive Pre-training

We conduct a preliminary study on contrastive pre-training and adapt PointContrast [18] to our problem setting with two schemes. For the first scheme, we positively pair the same objects with different rotation and Gaussian jitters and treat the remaining 2(N-1) samples in the mini-batch as negative examples. For the second scheme, we leverage the original point-level pairing and PointInfoNCE loss proposed in [18] as our pre-training objective. PointContrast leverages two different views of a point cloud scene and computes the point-level mapping for pairing. Such a construction procedure corresponds to rotation and scale transformations in their official implementation [19]. We empirically choose the temperature parameter = 0.07 in our experiments. Table B shows the results.

Table B: Evaluation Results (%) of PointContrast Pre-training.

| Pre-training Scheme | Parameters | ModelNet40 | | | | | | ScanObjectNN | | | | | | ModelNet10 | | | | | |
|---|---|---|---|---|---|---|---|---|---|---|---|---|---|---|---|---|---|---|---|
| | | PointNet | | DGCNN | | PCT | | PointNet | | DGCNN | | PCT | | PointNet | | DGCNN | | PCT | |
| | | CA | RA | CA | RA | CA | RA | CA | RA | CA | RA | CA | RA | CA | RA | CA | RA | CA | RA |
| AT Baseline | N/A | 87.7 | 37.9 | 90.6 | 62.0 | 89.7 | 49.1 | 69.9 | 23.7 | 74.4 | 30.9 | 72.4 | 20.5 | 96.6 | 79.7 | 98.1 | 86.3 | 97.4 | 80.0 |
| PointContrast | Adapted | 87.7 | 47.5 | 91.6 | 62.7 | 89.8 | 49.1 | 69.4 | 24.6 | 76.2 | 36.6 | 70.4 | 25.1 | 96.7 | 80.0 | 98.1 | 87.5 | 97.0 | 80.1 |
| PointContrast | Original | 87.2 | 47.0 | 91.5 | 62.5 | 89.7 | 49.1 | 69.5 | 24.3 | 75.6 | 32.8 | 70.5 | 22.9 | 96.9 | 79.5 | 98.0 | 86.4 | 97.1 | 80.1 |

We find that although our contrastive pre-training indeed shows improvements compared to the adversarial training (AT) baseline, it cannot beat the jigsaw-based pre-training strategy. We summarize several potential reasons. First, the current dataset for point cloud classification is much smaller than the datasets in 2D vision (e.g., ImageNet [20]). However, contrastive learning usually requires a large and sophisticated dataset to work well [21, 22]. In PointContrast, they need to rely on a much larger dataset, ScanNet [23] (consisting of both spatial and color information), even for the downstream classification task. Specifically, ScanNet contains 2.5 million views in more than 1,500 scans [23]. However, in our study, we find that pre-text task-based methods can work well purely on the original dataset. PointContrast also explicitly mentioned that even pre-training on ShapeNet [24], which is also extracted from CAD models but a much larger dataset than ModelNet40, cannot effectively improve the fine-tuning performance of point cloud learning. Second, to construct different views of a point cloud scene, PointContrast still leverages global transformation to pair points. Therefore, we believe such a scheme may also focus more on global feature learning. In an extreme case, if the network could learn the transformation applied in constructing different views, the network will accurately identify the positive pairs. Therefore, it will reduce to learning the transformation itself, which can explain why the results are no better than 3D rotation.

### B.2.3 Reproducible Results of Adversarial Pre-training for Fine-tuning

All the evaluation results from Table 2 in the main paper are based on experiments using the same random seed. To confirm the robustness improvements are non-trivial, we utilize five different random seeds to re-run the experiments. As it is too time-consuming and resource-intensive to re-schedule all the experiments for pre-training and fine-tuning, we only select the best adversarial pre-training for fine-tuning (APF) setting in Table 2 (main paper) for this evaluation. As Table C shows, we find that the variances of the fine-tuned models are relatively small and there are still significant improvements compared to the adversarial training baseline, which demonstrates that the pre-training on useful proxy tasks indeed help achieve stronger robustness in the fine-tuning stage.

### B.2.4 Evaluation of PGD Attack with Different $\epsilon$ and C&W Attack

We evaluate our adversarially trained model on more attacks including PGD attacks with $\epsilon \in [0.02, 0.04, 0.06]$ and C&W attack [25].

From Table D, we observe that models from our best APF strategy could achieve non-trivial robustness among different epsilons (even with $\epsilon = 0.06$), and it is expected that the adversarially trained models perform better with $\epsilon < 0.05$. Our models trained with self-supervised learning also achieve consistently stronger robustness than the AT baselines. It further verifies the significance of self-supervised learning for 3D point cloud robustness.

We also measure the attack success rate (ASR) of targeted C&W attack [25] with the $\ell_\infty$ norm constraint [26]. Table E shows that C&W attack cannot easily achieve the adversarial goal on models protected even by standard adversarial training.

Table C: Evaluation Results (%) of Adversarial Pre-training for Fine-tuning.

| | | ModelNet40 | | | | | | ScanObjectNN | | | | | | ModelNet10 | | | | | |
|---|---|---|---|---|---|---|---|---|---|---|---|---|---|---|---|---|---|---|---|
| | | PointNet | | DGCNN | | PCT | | PointNet | | DGCNN | | PCT | | PointNet | | DGCNN | | PCT | |
| | | CA | RA | CA | RA | CA | RA | CA | RA | CA | RA | CA | RA | CA | RA | CA | RA | CA | RA |
| AT Baseline | Average | 87.6 | 37.9 | 90.5 | 61.7 | 89.4 | 48.9 | 69.7 | 23.5 | 74.4 | 31.9 | 72.4 | 20.6 | 96.5 | 79.6 | 98.2 | 86.1 | 97.7 | 80.0 |
| | Variance | 0.15 | 3.76 | 0.15 | 1.01 | 0.29 | 0.82 | 0.23 | 1.71 | 0.13 | 0.98 | 0.38 | 0.53 | 0.08 | 0.23 | 0.05 | 0.19 | 0.07 | 0.21 |
| Best APF | Average | 87.9 | 51.8 | 90.7 | 66.9 | 89.9 | 51.0 | 70.6 | 25.6 | 76.2 | 40.8 | 73.1 | 27.2 | 97.0 | 80.7 | 98.3 | 90.5 | 97.3 | 83.8 |
| | Variance | 0.13 | 0.40 | 0.31 | 0.71 | 0.32 | 0.14 | 0.61 | 0.51 | 1.09 | 1.52 | 0.87 | 0.69 | 0.06 | 0.13 | 0.06 | 0.18 | 0.10 | 0.23 |

Table D: Robust Accuracy (%) of PGD Attack with Different $\epsilon$.

| Pre-training Scheme | Parameter | ModelNet40 | | | ScanObjectNN | | | ModelNet10 | | |
|---|---|---|---|---|---|---|---|---|---|---|
| | | PointNet | DGCNN | PCT | PointNet | DGCNN | PCT | PointNet | DGCNN | PCT |
| AT Baseline | $\epsilon = 0.02$ | 63.2 | 80.9 | 78.8 | 43.6 | 54.9 | 54.1 | 91.9 | 93.3 | 92.1 |
| Best APF | $\epsilon = 0.02$ | 76.0 | 84.3 | 79.4 | 47.4 | 62.0 | 56.1 | 92.1 | 96.2 | 94.6 |
| AT Baseline | $\epsilon = 0.04$ | 46.2 | 70.9 | 61.3 | 28.5 | 39.6 | 35.1 | 84.5 | 91.0 | 86.2 |
| Best APF | $\epsilon = 0.04$ | 61.1 | 76.7 | 62.9 | 31.8 | 48.9 | 37.4 | 85.2 | 92.9 | 88.4 |
| AT Baseline | $\epsilon = 0.06$ | 25.1 | 48.9 | 37.2 | 17.8 | 25.6 | 19.1 | 74.1 | 84.2 | 77.1 |
| Best APF | $\epsilon = 0.06$ | 43.2 | 54.1 | 39.6 | 19.3 | 33.1 | 20.5 | 75.0 | 86.4 | 77.6 |

Table E: Attack Success Rate (%) of C&W Attack.

| Pre-training Scheme | Parameter | ModelNet40 | | | ScanObjectNN | | | ModelNet10 | | |
|---|---|---|---|---|---|---|---|---|---|---|
| | | PointNet | DGCNN | PCT | PointNet | DGCNN | PCT | PointNet | DGCNN | PCT |
| ST Baseline | $\epsilon = 0.05$ | 98.3 | 98.1 | 99.0 | 100.0 | 100.0 | 100.0 | 95.3 | 96.2 | 95.1 |
| AT Baseline | $\epsilon = 0.05$ | 11.2 | 7.6 | 9.8 | 35.9 | 24.4 | 39.7 | 5.9 | 5.5 | 6.0 |
| Best APF | $\epsilon = 0.05$ | 6.9 | 5.2 | 5.7 | 30.1 | 20.0 | 30.4 | 5.5 | 4.2 | 5.5 |

### B.2.5 Transfer Attacks on ScanObjectNN and ModelNet10

Similarly, we demonstrate that fine-tuned models from different pre-training tasks have different vulnerabilities, and adversarial examples generated by attacking them do not transfer well among each other on ScanObjectNN and ModelNet10. As Figure D and E shows, there is still a ∼15% gap between robust accuracy on the diagonal and the other locations of the heat maps. We have presented the results of the ensemble methods in the main paper (Table 2).

### B.2.6 Point Dropping and Adding Threats on ModelNet

The quantitative results of point dropping (PD) and adding (PA) adversaries on ModelNet [27] are summarized in Table F and G. We find that jigsaw-based APF still achieves the highest robust accuracy, and DGCNN also performs the best among the three architectures. Since two ModelNet datasets are extracted from CAD models, modifications on a small number of points would not result in a large accuracy drop even in adversarial settings. We also observe that the robust accuracy will sometimes surpass the clean accuracy. The reason is that the adversarial training always takes the dropped or added point cloud samples as input so that the model fits the modified point clouds better, while the clean accuracy still maintains.

Table F: Evaluation Results (%) of PD on ModelNet40 and ModelNet10.

| Pretext Task | Parameters | ModelNet40 | | | | | | ModelNet10 | | | | | |
|---|---|---|---|---|---|---|---|---|---|---|---|---|---|
| | | PointNet | | DGCNN | | PCT | | PointNet | | DGCNN | | PCT | |
| | | CA | RA | CA | RA | CA | RA | CA | RA | CA | RA | CA | RA |
| ST Baseline | N/A | 88.6 | 66.6 | 91.5 | 75.5 | 92.1 | 72.2 | 96.0 | 79.0 | 97.8 | 89.0 | 97.9 | 88.3 |
| AT Baseline | N/A | 87.0 | 88.6 | 89.1 | 93.6 | 85.9 | 91.2 | 96.6 | 94.3 | 98.5 | 98.3 | 97.5 | 98.5 |
| 3D Jigsaw | $k = 3$ | 87.6 | 88.3 | 88.6 | 93.6 | 86.4 | 94.0 | 96.5 | 94.2 | 98.3 | 98.6 | 97.4 | 98.4 |
| | $k = 4$ | 86.7 | 88.7 | 87.6 | 94.3 | 85.4 | 94.3 | 97.0 | 94.7 | 98.5 | 98.5 | 97.3 | 98.4 |
| Adversarial 3D Jigsaw | $k = 3$ | 87.2 | 88.7 | 91.0 | 93.6 | 87.4 | 94.1 | 96.1 | 94.7 | 98.8 | 98.5 | 96.7 | 98.7 |
| | $k = 4$ | 87.4 | 89.1 | 90.1 | 93.4 | 87.5 | 94.2 | 96.8 | 95.0 | 97.8 | 98.4 | 98.0 | 98.8 |

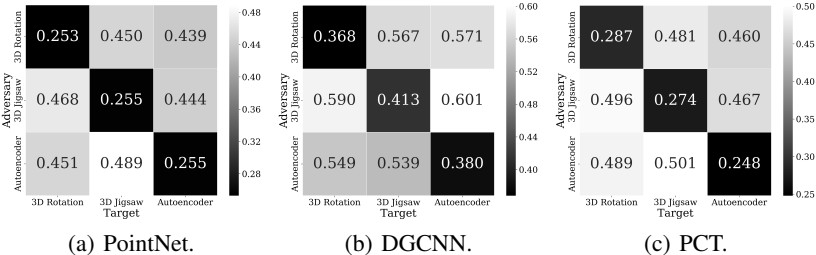

(a) PointNet.  (b) DGCNN.  (c) PCT.

Figure D: Robust Accuracy on Transfer Attacks among Fine-tuned Models from Different SSL Tasks on ScanObjectNN.

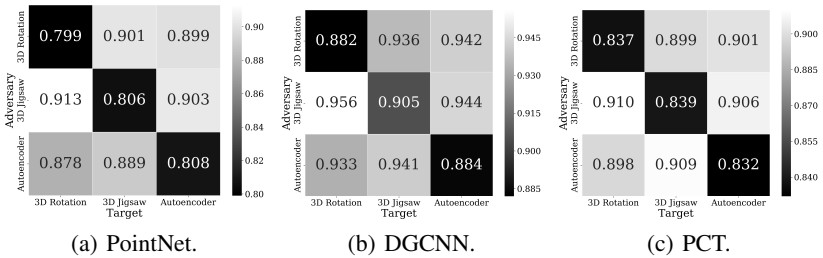

(a) PointNet.  (b) DGCNN.  (c) PCT.

Figure E: Robust Accuracy on Transfer Attacks among Fine-tuned Models from Different SSL Tasks on ModelNet10.

Table G: Evaluation Results (%) of PA on ModelNet40 and ModelNet10.

| | | ModelNet40 | | | | | | ModelNet10 | | | | | |
| | | PointNet | | DGCNN | | PCT | | PointNet | | DGCNN | | PCT | |
| Pretext Task | Parameters | CA | RA | CA | RA | CA | RA | CA | RA | CA | RA | CA | RA |
|---|---|---|---|---|---|---|---|---|---|---|---|---|---|
| ST Baseline | N/A | 88.6 | 78.6 | 91.5 | 65.5 | 92.1 | 52.6 | 96.0 | 95.2 | 97.8 | 93.1 | 97.9 | 93.1 |
| AT Baseline | N/A | 89.3 | 83.3 | 91.6 | 85.3 | 90.4 | 82.4 | 96.5 | 95.3 | 98.2 | 96.4 | 97.9 | 96.1 |
| 3D Jigsaw | $k = 3$ | 89.8 | 83.3 | 91.5 | 85.2 | 90.7 | 84.0 | 97.1 | 96.3 | 98.6 | 96.5 | 98.0 | 95.9 |
| | $k = 4$ | 89.6 | 83.3 | 91.7 | 85.5 | 90.0 | 82.2 | 96.8 | 95.3 | 98.7 | 95.9 | 97.9 | 95.0 |
| Adversarial | $k = 3$ | 89.5 | 83.4 | 91.4 | 85.7 | 90.0 | 85.0 | 96.8 | 95.8 | 98.6 | 96.5 | 98.1 | 96.6 |
| 3D Jigsaw | $k = 4$ | 89.0 | 83.4 | 92.2 | 86.6 | 90.4 | 85.1 | 96.6 | 95.4 | 98.1 | 96.5 | 97.9 | 96.4 |