# OpenReview forum: "Adversarially Robust 3D Point Cloud Recognition Using Self-Supervisions"
_NeurIPS.cc/2021/Conference — NeurIPS 2021 Poster_

### Official Review · Reviewer_CMgR · 2021-07-09

**Rating:** 5
**Confidence:** 4

**Summary:**

This paper presents studies on exploring the impact of various self-supervised learning tasks on three different architectures and three different threat models for 3D point clouds:
(1) three primary backbone architecture: a shared multi-layer-perceptron-based network PointNet, DGCNN, and a transformer-based network Point Cloud Transformer (PCT) [17];
(2) three threat models: point shifting (PS), point dropping (PD), and point adding (PA); and
(3) three self-supervised learning pretext task: 3D Rotation (predict 3D rotation angles), 3D Jigsaw, and Autoencoder (point cloud reconstruction).
Then, two strategies were used to integrate self-supervised learning and adversarial training, including (1) adversarial pre-training for fine-tuning (APF), which uses the SSL tasks only for pre-training, and (2) adversarial joint training (AJT), which jointly trains the SSL task with the recognition task.

Overall, the contribution of this paper is not on the method (Section 2) but on the experiments that compare different architectures, strategies, and models, and then show that what combinations will lead to better robustness.

In short, the experiment considers four datasets (ModelNet40, ModelNet10, ScanObjectNN, and ShapeNetPart) and two tasks: classification and segmentation.

**Ethical Concerns:**

Nil.

**Limitations And Societal Impact:**

Even though this is an exploratory paper that focuses on the experiments, the authors better discuss the limitations of the experiments.

**Main Review:**

Major weaknesses:

(1) The technical novelty is weak, as the contribution is more on the experiments and results rather than the method.

(2) For the threat models:
- From Figure 2, Point addition (PA) seems to be just adding random points near the surface, so the attack looks weak.
- Line 176 says that a larger number of modifications would be easy to detect. This argument sounds strange, cause this work is to explore how AT (adversarial training) can help to improve the robustness of the trained network, so dropping just 10% points does not sound sufficient.  It is better to perform the experiment by trying different (and larger) amount of point drops.
- The point drop is only 10%; doing so does not really affect global tasks, particularly classification. See Fig.5 in this paper: "SampleNet: Differentiable Point Cloud Sampling".  Dropping large proportion of points (>10%) still can't affect the classification accuracy.  Therefore, the setting for PD looks too weak, particularly for the classification experiment.
- Also, from Figure 2, it seems that all the salient points were on the bottom of the chair. Does it make sense? Why not consider existing methods for 3D key point detection?

(3) About the third architecture, i.e., PCT:
- The description on PCT [17] is too short and not informative in the paper. From supp., it seems that this PCT is developed based on EdgeConv (DGCNN) with k=32 (?).  Since this paper [17] is not published yet, should motivate why this is a good point transformer to pick, since there are other point transformers on arxiv, e.g., https://github.com/Strawberry-Eat-Mango/PCT_Pytorch and https://github.com/lucidrains/point-transformer-pytorch
- But in fact, why not also try other point transformers at the same time and see if you have consistent results?  The result in Section 3.1 seems to suggest that DGCNN is better than PCT [17], but I wonder if it is also true for other transformers (see above).

(4) I can see that the authors have tried their best to discuss the "insight" in each experiment but it is still very hard to understand the connections between the experimental results and the underlying reasons.

Overall, my rating is more borderline.

======================================================

Other comments:

Line 69: please use the full word for AT here. It was defined too early in line 33.

Line 177: what is the meaning of 14,5?  14 or 5?

Missing reference:
- Adversarial robustness: From self-supervised pre-training to fine-tuning. CVPR 2020.

**Time Spent Reviewing:**

4

---

> ### Author Response · Authors · 2021-08-10
> **Response to Reviewer CMgR [1/2]**
>
> We thank the reviewer for her/his constructive comments and agreeing that our work has covered multiple dimensions to analyze the robustness of 3D point cloud learning. Below we respond to each question of the reviewer:
>
> > Q: The technical novelty is weak, as the contribution is more on the experiments and results rather than the method.
>
> A : We believe that our work is novel for the following reasons:
>
> 1. From the problem definition side, we are the first to study the relationship between self-supervised learning and adversarial robustness in the 3D space, which was largely unexplored. This is a new and important research problem. It contains essential differences in studying the relationship between 3D adversarial robustness and 3D SSL compared to the 2D domain. For instance, different from prior work in adversarial robustness in 2D vision that either leverages a single SSL task [1] or targets a single backbone model [2], 3D point cloud recognition using deep learning is a relatively nascent field, where finding proper universal “3D backbones” is still an active research direction. Understanding which 3d backbone is more robust in the early stage could help guide the community in the right direction. Therefore, it is necessary to consider different types of learning architectures to study their robustness. Moreover, given the intrinsic sparsity and set property of 3D point cloud data, various 3D point cloud domain-specific attack threat models (e.g. point shifting, point adding, and point dropping) should be investigated. Therefore, in this paper, we conduct a much more comprehensive study along multiple 3D domain-specific dimensions to study its robustness including three 3D SSL tasks, three different types of attack threats, and three backbone architectures based on the properties of point cloud data. It is also worth noting that we, for the first time, formulate point adding (PA) and dropping (PD) adversaries into a general AT analysis framework. As also mentioned by the reviewer, we believe such a study could be useful to the community and future research in this domain.
>
> 2. From the technical perspective, improving the adversarial robustness of 3D point clouds with self-supervised learning tasks is a non-trivial problem. We find that while some designs in 3D point cloud models work well in standard training, but they fail to provide robustness in adversarial training. The reason is that these designs introduce **randomness** in the inner maximization stage of adversarial training, causing the overall training intractable. For example, as illustrated in Section 3.2, we find that several components in PointNet and PCT (e.g., T-Net and farthest point sampling (FPS)) cannot work well by directly applying adversarial training. T-Net applies an overall transformation matrix to the point clouds and middle layer features, where a small perturbation in the matrix could affect all the features. Therefore, T-Net will introduce a high variance of performance in adversarial training. PCT originally leverages farthest point sampling (FPS) to sample anchors to do local clustering. Since the point cloud is dynamically changing in adversarial training, the sampled anchors are totally different in each PGD iteration, which will make models confusing. Therefore, the trained model is still not robust. We apply multiple modifications, including replacing the FPS operation with EdgeConv to stabilize the adversarial training (detailed in Section 3.2 and Appendix B). The capacity of EdgeConv is a superset of FPS operation since it views every point as an anchor and performs clustering so that it removes the randomness introduced by FPS. **Such a change does not modify the usage of the transformer in PCT, as shown in Appendix B, and will improve both clean and robust accuracy for the baseline.**
>
> 3. From the experimental analysis side, our analysis unveils multiple insights which are unique in 3D point cloud learning. For example, [2] shows that jigsaw SSL does not perform as well as rotation in 2D vision for robustness enhancement. However, 3D jigsaw helps connect the global and local feature learning, which improves a lot in 3D adversarial training. We also take the first step to analyze the robustness in the part segmentation task and provide valuable insights.
>
> We believe a systematic study with insights should be considered a more critical factor, especially in the adversarial machine learning community, since fancier solutions may give a false sense of security that would be adaptively broken [3]. Our study further motivates future research on designing more robust operations, architectures, and SSL tasks in 3D vision.
>
>
> [1] Hendrycks, Dan, et al. "Using Self-Supervised Learning Can Improve Model Robustness and Uncertainty." Advances in Neural Information Processing Systems 32 (2019): 15663-15674.
>
> [2] Chen, Tianlong, et al. "Adversarial robustness: From self-supervised pre-training to fine-tuning." Proceedings of the IEEE/CVF Conference on Computer Vision and Pattern Recognition. 2020.
>
> [3] Athalye, Anish, Nicholas Carlini, and David Wagner. "Obfuscated gradients give a false sense of security: Circumventing defenses to adversarial examples." International conference on machine learning. PMLR, 2018.
>
> > Q: Point addition (PA) seems to be just adding random points near the surface, so the attack looks weak.
>
> A: We appreciate the reviewer for carefully reviewing our formulations. However, point addition (PA) is NOT just adding random points near the surface. We need to run the adversarial attacks after initiating the added points. In our study, we follow two principles to set up the threat model. First, the perturbation will not affect human perception much. Second, the perturbation is more than effective to break the clean trained models.
> Adding or dropping a large amount of points will also have a major effect on human perception [1,2]. In the published paper that proposes PA attack [1], they suggest adding 50 to 100 points bounding by informal Chamfer distance. Therefore, we adopt their experimental setting to make sure that our threat model will not affect human perception.
>
> [1] Xiang, Chong, Charles R. Qi, and Bo Li. "Generating 3d adversarial point clouds." Proceedings of the IEEE/CVF Conference on Computer Vision and Pattern Recognition. 2019.
>
> [2] Zheng, Tianhang, et al. "Pointcloud saliency maps." Proceedings of the IEEE/CVF International Conference on Computer Vision. 2019.
>
> > Q: Point dropping attack is not effective.
>
> A: We believe there are some misunderstandings for the point dropping attack. The reviewer is right that by randomly sampling the point cloud, even dropping 30% of the points will not have a major impact on the classification accuracy. The goal of the referenced work [1] is to maintain the classification accuracy by sampling the point cloud. However, our point dropping is conducted in an **adversarial** manner using the method proposed in [2]. The adversary tries to drop the most influential points with regard to the classification accuracy. [2] shows that with 200 dropped points, the remaining point cloud can even fool human perception. Therefore, we restrict the threat model to be capable of dropping 100 points to not affect human perception of the remaining point clouds.
>
> [1] Lang, Itai, Asaf Manor, and Shai Avidan. "Samplenet: Differentiable point cloud sampling." Proceedings of the IEEE/CVF Conference on Computer Vision and Pattern Recognition. 2020.
>
> [2] Zheng, Tianhang, et al. "Pointcloud saliency maps." Proceedings of the IEEE/CVF International Conference on Computer Vision. 2019.
>
> > Q: The dropped points were on the bottom of the chair.
>
> A: This is actually a piece of good evidence and complement for the above question. As we mentioned, the point dropping is done in an **adversarial** manner. We leverage the saliency point detection method in [1] to drop the most influential points to decrease the classification accuracy. In this example, the adversarial optimization finds that the most salient points are at the bottom of the chair to change its predicted class, which is also somewhat consistent with human perception. Figure 8 in [1] also shows a similar example. We hope these two answers will eliminate the concerns of the point dropping threat model.
>
> [1] Zheng, Tianhang, et al. "Pointcloud saliency maps." Proceedings of the IEEE/CVF International Conference on Computer Vision. 2019.

---

> > ### Author Response · Authors · 2021-08-10
> > **Response to Reviewer CMgR [2/2]**
> >
> > > Q: Question about the adopted PCT architecture.
> >
> > A: Due to space constraints, we put the detailed layer information of each adopted architecture (i.e., PointNet, DGCNN, and PCT) in the appendix. We will revise our manuscript to add them to the main paper.
> >
> > As the reviewer indicated, there are two state-of-the-art transformer-based point cloud recognition models: PCT and PointTransformer.
> > We indeed adopt the implementation of PCT: “https://github.com/Strawberry-Eat-Mango/PCT_Pytorch” in our study.
> > We have also tried PointTransformer: “https://github.com/lucidrains/point-transformer-pytorch” at the beginning of our study. However, the default implementation requires the normal direction of each point, which is additional information to the location of each point (i.e., xyz). To have a fair comparison, we have tried to train it without normal information and find that the test-time clean accuracy can only achieve around 81% which is even much worse than PointNet's 87.1%. It is worth mentioning that the goal of our study is not to design a new transformer architecture. Therefore, we didn’t choose the PointTransformer in our study.
> >
> > We only change two layers inside PCT from the PointNet++ operation [2] to EdgeConv, and we have introduced the detailed reason in Section 3.2: the PointNet++ operation leverages farthest point sampling (FPS) to sample anchors to do clustering. Since the point cloud is dynamically changing in adversarial training, FPS will make adversarial training unstable and make the trained model not robust. The capacity of EdgeConv is a superset of PointNet++ operation since it views every point as an anchor and performs clustering so that it removes the randomness introduced by FPS. **Such a change does not modify the usage of the transformer in PCT, as shown in Appendix B, and will improve both clean and robust accuracy for the baseline.**
> >
> > Since transformer-based 3D point cloud recognition is a nascent field, we also would like to use our study as motivation for future studies to design more robust transformer architectures for 3D point cloud learning.
> >
> > > Q: Connection between experimental results and insights.
> >
> > A: As the reviewer indicated, we have tried our best to explain our experiments' underlying reasons and provide useful insights. We have included an "insight" section in each experiment section to provide useful information for the community and motivate future research. For example, through our experiments, we cross-validate the same characteristics from DGCNN and the jigsaw pre-text task, which is to strengthen the connection between local and global feature learning. We welcome suggestions from the reviewer to make our manuscript more fluent and clear.
> >
> > > Q: What does “k = 14,5” means in Line 177
> >
> > A: We apologize for this confusing notation. The reviewer is correct that we leverage k = 14 in the training phase and k = 5 in the testing phase. The rationale is that we would like to leverage a more severe attacker in the testing phase to evaluate the true robust accuracy. With k = 5, we will have N / k = 100 / 5 = 20 iterations to perturb a point cloud, which is more fine-grained than 7 iterations in the training time; this will give us the robust accuracy closer to the lower bound.
> >
> > We do include the reference in our paper, which is [25].
> >
> > We will fix the grammar issues in our revised manuscript. We believe that we have addressed all the raised concerns, and we are happy to answer any follow-up questions. We hope the reviewer might reconsider the rating based on our response.

---

### Official Review · Reviewer_Dx1d · 2021-07-11

**Rating:** 6
**Confidence:** 3

**Summary:**

The paper systematically studies the impact of various self-supervised learning proxy tasks on different architectures and threat models for 3D point clouds. Experiments on three datasets imply that self-supervised learning (SSL), as a pretraining task, improves the robustness of the 3D point-cloud-based model for classification. It is observed that DGCNN and the 3D jigsaw SSL task are more effective w.r.t. robustness, which is claimed to benefit from more robust local features.

**Limitations And Societal Impact:**

The authors have not adequately addressed the limitations.

**Main Review:**

Strongness:
+ Comprehensive experiments on multiple datasets
+ The overall message is clear: DGCNN and the 3D jigsaw SSL task are more effective w.r.t. robustness.

---

Weakness and Questions:
1. The clarity of the paper can be improved.
    - L164: $D_t$ is not well defined yet. I assume that it is a data point indexed by $t$.
    - There are many abbreviations, e.g. ST, AT, CA, etc. It is better to highlight those words where they are defined, so that the readers are able to quickly localize their meanings when they read the experiment section. Alternatively, the definitions of those abbreviations can be emphasized again in the table or figure captions. Besides, the definition of RA seems to be missing, while I assume that it stands for the accuracy after the attack.
    - (minor) The overview paragraph of each section is not informative enough currently. Although it actually describes the following content, it fails to guide me when I first read it.
2. From Table 2, it seems that the impact of SSL (self-supervised learning) and AT (adversarial training) are not complementary. For example, AT + SSL (jigsaw) seems to perform similarly or even worse than SSL only. Can the authors explain it?
3. Is there any data augmentation in ST and AT baseline? If no, can the authors add baselines with data augmentation similar to your SSL tasks (rotation and jigsaw)?
4. It is questionable whether the contribution of the paper meets the standard of NeurIPS. From my perspective, the paper focuses on scientific observations by comparing different existing network architectures and pretext tasks w.r.t robustness. According to the related works of the paper, there are no significant technical contributions. However, the observations proposed by this paper are interesting: learning robust local features are important. Despite the comparison with PointNet and PCT, the lack of deeper analysis on local features is a drawback. For example, a common practice of SSL is contrastive learning, and there are several works about 3D SSL (like PointContrast), which also focus on local feature learning. More pretext tasks relevant to local feature learning can be included to verify the point. Besides, the autoencoder (shape reconstruction) can also rely on local features, like NSFA(Detail preserved point cloud completion via separated feature aggregation.), which can provide an apple-to-apple comparison about the impact of local feature learning.

---

Typo:
L201: default -> by default

---
**Update after rebuttal**
According to the additional results presented in the rebuttal, I slightly raise my score to 6 (Marginally above the acceptance threshold), and hope to see the additional results are included in the revision.



**Time Spent Reviewing:**

3

---

> ### Author Response · Authors · 2021-08-10
> **Response to Reviewer Dx1d [1/2]**
>
> We thank the reviewer for her/his constructive comments and agreeing that our work has presented a comprehensive study and strong empirical insights: DGCNN and 3D jigsaw perform better in adversarial robustness. Below we respond to each question of the reviewer:
>
> > Q: The clarity of the paper can be improved.
>
> A: We agree with the reviewer that the organization of our manuscript can be further improved.
> $D_{t}$ denotes the data for the self-supervised learning task $t$. Yes, CA and RA denote clean accuracy and robust accuracy, respectively. We will revise our paper to make it more clear.
>
> > Q: From Table 2, it seems that the impact of SSL (self-supervised learning) and AT (adversarial training) are not complementary.
>
> A: There are indeed some misunderstandings. There are two types of adversarial training in our adversarial pre-training for fine-tuning (APF) study: (1) **adversarial pre-training** for the SSL task and (2) **adversarial fine-tuning** for the downstream classification task. “Adversarial X” means that we additionally use **adversarially pre-training** for the self-supervised learning (SSL) task X, and “X” means that we only use **clean pre-training** for SSL task X. All the results in Table 2 are applied with  **adversarial fine-tuning** for the ultimate classification task. In fact,  all of our results, including “Adversarial X”  and “X” in Table 2, actually reflect the performance of SSL + adversarial training. Both “Adversarial X”  and “X” show better performance than baseline (adversarial training) so that our observations and insights that SSL is essential to the robustness improvements still hold.
>
> By adversarially pre-training the SSL task (e.g., rotation), the models will be arguably more robust against SSL task-specific attacks (e.g., adversarial rotation attack). We study the  **adversarial pre-training** +  **adversarial fine-tuning** setting because we want to observe whether such robustness against SSL task-specific attacks (e.g. adversarial rotation attack)  might transfer to the backbone for the downstream supervised classification task with  **adversarial fine-tuning**. However, this hypothesis is not always true. Therefore, we get a more accurate conclusion that  **adversarial fine-tuning** is the key to the robustness improvements.
>
> > Q: Is there any data augmentation in standard training (ST) and adversarial training (AT) baseline?
>
> A: We leverage Gaussian jitter to augment the ST and AT baseline. We follow the reviewer’s suggestions to add rotation and jigsaw augmentation to test the performance on ModelNet40.
>
> |     Rotation Augmentation        || ModelNet40            |           |
> |:----:|:----:|:----:|:----:|
> | CA/RA (%)   | PointNet   | DGCNN     | PCT       |
> | ST Baseline | 85.2/0     | 90.9/0    | 90.1/0    |
> | AT Baseline | 82.2/17.5  | 87.2/49.2 | 85.1/38.1 |
>
> |      Jigsaw Augmentation       || ModelNet40         |        |
> |:----:|:----:|:----:|:----:|
> | CA/RA (%)   | PointNet   | DGCNN  | PCT    |
> | ST Baseline | 2.41/0     | 2.66/0 | 2.42/0 |
> | AT Baseline | 2.43/0     | 2.56/0 | 2.49/0 |
>
> We find that both augmentations will hurt the performance, and there are some reasons.
>
> 1. Although rotated point clouds preserve the global shape with the original point clouds, the designs of the point cloud recognition models are not rotation-invariant. Therefore, rotation augmentation will hurt the performance, since the model has limited ability to adapt the rotated data.
>
> 2. Jigsaw itself does not fit as an augmentation method since it will displace the point cloud to different locations which will make the data totally different from the original distribution, as shown in Figure 1. Therefore, in the test time, it is not possible for the model to predict the original point cloud’s class. It explains that with jigsaw augmentation, the model’s decision becomes like a random guess.
>
> **It is worth mentioning that self-supervised learning (SSL) is fundamentally different from data augmentation.** Our SSL is to make the model predict the transformation itself to learn robust context information. However, data augmentation is to make the model generalize to different types of data.
>
> > Q: There are few significant technical contributions
>
> A: We believe that we have significant contributions.
>
> 1. From the problem definition side, we are the first to study the relationship between self-supervised learning and adversarial robustness in the 3D space, which was largely unexplored. This is a new and important research problem. It contains essential differences in studying the relationship between 3D adversarial robustness and 3D SSL compared to the 2D domain. For instance, different from prior work in adversarial robustness in 2D vision that either leverages a single SSL task [1] or targets a single backbone model [2], 3D point cloud recognition using deep learning is a relatively nascent field, where finding proper universal “3D backbones” is still an active research direction. Understanding which 3d backbone is more robust in the early stage could help guide the community towards the right direction. Therefore, it is necessary to consider different types of learning architectures to study their robustness. Moreover, given the intrinsic sparsity and set property of 3D point cloud data, various 3D point cloud domain-specific attack threat models (e.g. point shifting, point adding, and point dropping) should be investigated. Therefore, in this paper, we conduct a much more comprehensive study along multiple 3D domain-specific dimensions to study its robustness including three 3D SSL tasks, three different types of attack threats, and three backbone architectures based on the properties of point cloud data. It is also worth noting that we, for the first time, formulate point adding (PA) and dropping (PD) adversaries into a general AT analysis framework. As also mentioned by the reviewer, we believe such a study could be useful to the community and future research in this domain.
>
> 2. From the technical perspective, improving the adversarial robustness of 3D point clouds with self-supervised learning tasks is a non-trivial problem. We find that while some designs in 3D point cloud models work well in standard training, but they fail to provide robustness in adversarial training. The reason is that these designs introduce **randomness** in the inner maximization stage of adversarial training, causing the overall training intractable. For example, as illustrated in Section 3.2, we find that several components in PointNet and PCT (e.g., T-Net and farthest point sampling (FPS)) cannot work well by directly applying adversarial training. T-Net applies an overall transformation matrix to the point clouds and middle layer features, where a small perturbation in the matrix could affect all the features. Therefore, T-Net will introduce a high variance of performance in adversarial training. PCT originally leverages farthest point sampling (FPS) to sample anchors to do local clustering. Since the point cloud is dynamically changing in adversarial training, the sampled anchors are totally different in each PGD iteration, which will make models confusing. Therefore, the trained model is still not robust. We apply multiple modifications, including replacing the FPS operation with EdgeConv to stabilize the adversarial training (detailed in Section 3.2 and Appendix B). The capacity of EdgeConv is a superset of FPS operation since it views every point as an anchor and performs clustering so that it removes the randomness introduced by FPS. **Such a change does not modify the usage of the transformer in PCT, as shown in Appendix B, and will improve both clean and robust accuracy for the baseline.**
>
> 3. From the experimental analysis side, our analysis unveils multiple insights which are unique in 3D point cloud learning. For example, [2] shows that jigsaw SSL does not perform as well as rotation in 2D vision for robustness enhancement. However, 3D jigsaw helps connect the global and local feature learning, which improves a lot in 3D adversarial training. We also take the first step to analyze the robustness in the part segmentation task and provide valuable insights.
>
> We believe a systematic study with insights should be considered a more critical factor, especially in the adversarial machine learning community, since fancier solutions may give a false sense of security that would be adaptively broken [3]. Our study further motivates future research on designing more robust operations, architectures, and SSL tasks in 3D vision.
>
>
> [1] Hendrycks, Dan, et al. "Using Self-Supervised Learning Can Improve Model Robustness and Uncertainty." Advances in Neural Information Processing Systems 32 (2019): 15663-15674.
>
> [2] Chen, Tianlong, et al. "Adversarial robustness: From self-supervised pre-training to fine-tuning." Proceedings of the IEEE/CVF Conference on Computer Vision and Pattern Recognition. 2020.
>
> [3] Athalye, Anish, Nicholas Carlini, and David Wagner. "Obfuscated gradients give a false sense of security: Circumventing defenses to adversarial examples." International conference on machine learning. PMLR, 2018.

---

> > ### Author Response · Authors · 2021-08-10
> > **Response to Reviewer Dx1d [2/2]**
> >
> > > Q: Analysis of contrastive learning is preferred.
> >
> > A: We adapt PointContrast [1] to our problem setting. Our designed scheme is as follows:
> > We positively pair the same objects with different rotation and Gaussian jitter augmentations and treat the remaining 2(N-1) samples in the mini-batch as negative examples. We attach an MLP head to the backbone to extract latent features for constructing the similarity matrix. We empirically choose the temperature parameter = 0.07 in our experiments. The following table shows the results.
> >
> >
> > |           | ModelNet40 |           |           | ScanObjectNN |           |           | ModelNet10 |           |           |
> > |    :----:   |:----:   |:----:   |:----:   |:----:   |:----:   |:----:   |:----:   |:----:   |:----:   |
> > | CA/RA (%) | PointNet   | DGCNN     | PCT       | PointNet     | DGCNN     | PCT       | PointNet   | DGCNN     | PCT       |
> > |           | 87.7/47.5  | 91.6/62.7 | 89.8/49.1 | 69.4/24.6    | 76.2/36.6 | 70.4/25.1 | 96.7/80.0  | 98.1/87.5 | 97.0/80.1 |
> >
> > We find that although our contrastive pre-training indeed shows improvements compared to the adversarial training (AT) baseline, it cannot beat the jigsaw-based pre-training strategy. We summarize two potential reasons. First, the current dataset for point cloud classification is much smaller than the datasets in 2D vision (e.g., ImageNet). However, contrastive learning usually requires a large dataset to work well [2,3,4]. In PointContrast, they need to rely on a much larger dataset, ScanNet [5] (consisting of both spatial and color information), even for the downstream classification task (the classification task is done on other datasets). However, in our study, we find that pre-text task-based methods can work well purely on the original dataset. Second, a contrastive learning design for point cloud classification is desired since PointContrast is more useful for the segmentation task. We will revise our manuscript to add this experiment and discuss the reasons. We also propose contrastive learning for adversarial robustness as a promising research direction to be followed up on by making improvements such as using a larger dataset.
> >
> > [1] Xie, Saining, et al. "Pointcontrast: Unsupervised pre-training for 3d point cloud understanding." European Conference on Computer Vision. Springer, Cham, 2020.
> >
> > [2] Chen, Ting, et al. "A simple framework for contrastive learning of visual representations." International conference on machine learning. PMLR, 2020.
> >
> > [3] Cao, Yun-Hao, and Jianxin Wu. "Rethinking Self-Supervised Learning: Small is Beautiful." arXiv preprint arXiv:2103.13559 (2021).
> >
> > [4] He, Kaiming, et al. "Momentum contrast for unsupervised visual representation learning." Proceedings of the IEEE/CVF Conference on Computer Vision and Pattern Recognition. 2020.
> >
> >
> > > Q: The autoencoder task can also focus on the local feature learning.
> >
> > A: We thank the reviewer for pointing out related work. However, we find [1] and other state-of-the-art autoencoder studies in 3D point clouds only cover a small subset of the classes in the ShapeNet dataset [2] (e.g., chair and car). During our experimentations on FoldingNet, we find that if we feed the complete ModelNet40 dataset into the training process, the autoencoder can only reconstruct a very rough shape of the original point cloud. We believe it will be too hard for the autoencoder to reconstruct the details for objects in all 40 classes. We do agree with the reviewer that this is a promising direction for designing more general autoencoders for better local feature learning. We will include this discussion in the revised version of our manuscript.
> >
> > [1] Zhang, Wenxiao, Qingan Yan, and Chunxia Xiao. "Detail preserved point cloud completion via separated feature aggregation." Computer Vision–ECCV 2020: 16th European Conference, Glasgow, UK, August 23–28, 2020, Proceedings, Part XXV 16. Springer International Publishing, 2020.
> >
> > [2] https://paperswithcode.com/task/point-cloud-generation/codeless
> >
> > We will fix the grammar issues in our revised manuscript. We believe that we have addressed all the raised concerns, and we are happy to answer any follow-up questions. We hope the reviewer might reconsider the rating based on our response.

---

> > > ### Comment · Reviewer_Dx1d · 2021-08-15
> > > **Questions related to PointContrast**
> > >
> > > Thanks for providing the experiments related to contrastive learning. However, the "adaptation" of PointContrast seems to deviate from the objective to show that "learning robust local features are important", since the adaption makes the SSL task close to "3D rotation". I think it is somehow necessary to have two SSL tasks to show that "learning robust local features are important" for adversarial robustness. The original PointContrast paper conducts SSL on point-level rather than object-level (or more precisely scene-level), which is more suitable than 3D rotation.
> > >
> > > As mentioned by most reviewers, the "technical" (not scientific) contribution is limited while the observations might give insights to the community. The design choice of using T-Net or FPS is more about a trade-off according to applications (e.g., most segmentation tasks do not include T-Net already, or they will use PointNet++ or other stronger backbones. Downsample is required to improve efficiency, while random downsampling might also be used (RandLA-Net).) Thus, I suggest in the original review that analyzing the impact of local feature learning might increase the depth of the paper.

---

> > > > ### Author Response · Authors · 2021-08-30
> > > > **Response to Reviewer Dx1d**
> > > >
> > > > We thank the reviewer for the detailed comment. We follow the suggestion from the reviewer to strictly use the experimental setting in PointContrast [1] in our study. Specifically, we leverage the point-level pairing and PointInfoNCE loss proposed in [1] as our pre-training objective. PointContrast leverages two different views of a point cloud scene and computes the point-level mapping for pairing. Such a construction procedure corresponds to rotation and scale transformations in their codebase implementation. We adopt their construction procedure as well. However, our evaluation results show that its performance is no better than our 3D rotation self-supervised learning setups.
> > > >
> > > > We believe multiple reasons can explain the results. 1. As we mentioned, contrastive learning usually requires a large and **sophisticated** dataset to work well. PointContrast relies on a much larger dataset, ScanNet [2] (consisting of both spatial and color information), than ModelNet and ScanObjectNN. Specifically, ScanNet contains 2.5 million views in more than 1500 scans [2]. In contrast, the largest dataset in our study, ModelNet40, only has around 10,000 samples. In PointContrast, the authors also explicitly mentioned that even pre-training on ShapeNet, which is also extracted from CAD models but is a much larger dataset than ModelNet40, cannot effectively improve the fine-tuning performance of point cloud learning. 2. As the reviewer pointed out, to construct different views of a point cloud scene, PointContrast still leverages **global** transformation to pair points. Therefore, we believe such a scheme may also focus more on global feature learning. In an extreme case, if the network can learn the transformation applied in constructing different views, the network will accurately identify the positive pairs. Therefore, it will reduce to learning the transformation itself, which can explain why the results are no better than 3D rotation.
> > > >
> > > > We agree that a local feature-focused contrastive learning scheme in point cloud recognition is desired in this community. We will discuss this in our revised paper and set it as our future work.
> > > >
> > > > We thank the reviewer for acknowledging that our observations are insightful. We agree that those design choices of using T-Net or FPS will also trade-off robustness. Therefore, our study, for the first time, unveiled their robustness characteristics under adversarial training. We believe it is beneficial to future research in this area.

---

### Official Review · Reviewer_ygsD · 2021-07-12

**Rating:** 7
**Confidence:** 5

**Summary:**

This work builds on recent adversarial attacks on 3D point clouds and recent self-supervised methods on 3D ( 3D jigsaw[28] and 3D rotation prediction[27]) to evaluate the robustness of the 3D models that were trained in such SSL setup. Comprehensive experiments are conducted on 3 different benchmarks in 3D classification and one in 3D part segmentation that show a marginal improvement in robustness by adding SSL pre-training before performing a standard adversarial training. Finally, the authors perform an ensemble of different robust models to improve the robustness of 3D point cloud classification.

**Limitations And Societal Impact:**

Adding a section about the vulnerabilities of 3D deep learning methods in safety-critical applications ( like self-driving cars systems) would be appreciated.

**Main Review:**

**Strengths** :

- The work provides comprehensive 3D classification experiments on several 3D datasets, including the realistic ScanObjectNN.

- The first work to investigate the effect of self-supervision on 3D deep learning robustness, as well as studying the robustness of 3D segmentation.


**Weaknesses**:

- The paper lacks technical novelty. All the SSL and adversarial attacks and defenses are developed in previous works, which makes the manuscript more or less an evaluation paper.  The proposed Joint SSL adversarial Training (AJT) is shown in the paper to be less effective than the naive SSL pre-training (APF). Performing an ensemble of robust models is not novel since it was shown before in [b]

- One of the findings of the paper ( that DGCNN is more robust than other architectures) was already found in [73] in their comprehensive study of different 3D architectures, including DGCNN.

- The work has a flawed experimentation setup. While the authors did show improvement in robustness, there are some caveats in the setup. The authors used a single $\epsilon$ value for the PGD attack ( 0.005) based on a rejected ICLR’21 submission [21]. Published works (like [12,19,73]) usually report several $\epsilon$ values for PGD attack or run on C&W formulation to make sure that the defense/attack methods work on generic attacks and not for that specific $\epsilon$ ( that might be too small for the attack to work).

- The uncertainty in the results is not quantified. It is not clear whether the small improvements over AT baseline ( that can be as low as 1%) are statistically significant. Reporting the variance of the results is highly appreciated in this scenario.



**Minor issues** :

- Some grammar mistakes need to be fixed ( line 117 e.g. )

- The paper misses recent work on 3D attacks [a], and ensemble defenses [b].

- The presentation of the paper can be improved. For example, Table 2  has 323 reported results, which is very difficult to follow.  Using plots, charts, and other visualizations can improve the presentation of the paper. The text size should be consistent in all of the figures and plots. Fig 3 has tiny legends for example.



[a] Tsai et.al, Robust adversarial objects against deep learning models. AAAI Conference on Artificial Intelligence (2020)

[b]Florina et.al,  Ensemble Adversarial Training: Attacks and Defenses, ICLR (2018)


**Time Spent Reviewing:**

4

---

> ### Author Response · Authors · 2021-08-10
> **Response to Reviewer ygsD [1/2]**
>
> We thank the reviewer for her/his constructive comments and agreeing that our work presents a comprehensive analysis on the robustness of 3D point clouds and we are the first to investigate the connection between self-supervised learning and robustness in 3D. Below we respond to each question of the reviewer:
>
> > Q: The novelty of the proposed strategy is limited.
>
> A: We believe that our study is novel due to the following reasons.
>
> 1. From the problem definition side, we are the first to study the relationship between self-supervised learning and adversarial robustness in the 3D space, which was largely unexplored. This is a new and important research problem. It contains essential differences in studying the relationship between 3D adversarial robustness and 3D SSL compared to the 2D domain. For instance, different from prior work in adversarial robustness in 2D vision that either leverages a single SSL task [1] or targets a single backbone model [2], 3D point cloud recognition using deep learning is a relatively nascent field, where finding proper universal “3D backbones” is still an active research direction. Understanding which 3d backbone is more robust in the early stage could help guide the community towards the right direction. Therefore, it is necessary to consider different types of learning architectures to study their robustness. Moreover, given the intrinsic sparsity and set property of 3D point cloud data, various 3D point cloud domain-specific attack threat models (e.g. point shifting, point adding, and point dropping) should be investigated. Therefore, in this paper, we conduct a much more comprehensive study along multiple 3D domain-specific dimensions to study its robustness including three 3D SSL tasks, three different types of attack threats, and three backbone architectures based on the properties of point cloud data. It is also worth noting that we, for the first time, formulate point adding (PA) and dropping (PD) adversaries into a general AT analysis framework. As also mentioned by the reviewer, we believe such a study could be useful to the community and future research in this domain.
>
> 2. From the technical perspective, improving the adversarial robustness of 3D point clouds with self-supervised learning tasks is a non-trivial problem. We find that while some designs in 3D point cloud models work well in standard training, but they fail to provide robustness in adversarial training. The reason is that these designs introduce **randomness** in the inner maximization stage of adversarial training, causing the overall training intractable. For example, as illustrated in Section 3.2, we find that several components in PointNet and PCT (e.g., T-Net and farthest point sampling (FPS)) cannot work well by directly applying adversarial training. T-Net applies an overall transformation matrix to the point clouds and middle layer features, where a small perturbation in the matrix could affect all the features. Therefore, T-Net will introduce a high variance of performance in adversarial training. PCT originally leverages farthest point sampling (FPS) to sample anchors to do local clustering. Since the point cloud is dynamically changing in adversarial training, the sampled anchors are totally different in each PGD iteration, which will make models confusing. Therefore, the trained model is still not robust. We apply multiple modifications, including replacing the FPS operation with EdgeConv to stabilize the adversarial training (detailed in Section 3.2 and Appendix B). The capacity of EdgeConv is a superset of FPS operation since it views every point as an anchor and performs clustering so that it removes the randomness introduced by FPS. **Such a change does not modify the usage of the transformer in PCT, as shown in Appendix B, and will improve both clean and robust accuracy for the baseline.**
>
> 3. From the experimental analysis side, our analysis unveils multiple insights which are unique in 3D point cloud learning. For example, [2] shows that jigsaw SSL does not perform as well as rotation in 2D vision for robustness enhancement. However, 3D jigsaw helps connect the global and local feature learning, which improves a lot in 3D adversarial training. We also take the first step to analyze the robustness in the part segmentation task and provide valuable insights.
>
> 4. Our ensemble methods are based on our transferability analysis of different adversarially fine-tuned models that preserve different vulnerabilities. Therefore, simple but effective ensemble methods can tangibly improve the robustness. In contrast, [4] leverages different attacks/threats to adversarially train the ensemble model. We believe that our study highlights different insights from [4].
>
> We believe a systematic study with insights should be considered a more critical factor, especially in the adversarial machine learning community, since fancier solutions may give a false sense of security that would be adaptively broken [3]. Our study further motivates future research on designing more robust operations, architectures, and SSL tasks in 3D vision.
>
>
> [1] Hendrycks, Dan, et al. "Using Self-Supervised Learning Can Improve Model Robustness and Uncertainty." Advances in Neural Information Processing Systems 32 (2019): 15663-15674.
>
> [2] Chen, Tianlong, et al. "Adversarial robustness: From self-supervised pre-training to fine-tuning." Proceedings of the IEEE/CVF Conference on Computer Vision and Pattern Recognition. 2020.
>
> [3] Athalye, Anish, Nicholas Carlini, and David Wagner. "Obfuscated gradients give a false sense of security: Circumventing defenses to adversarial examples." International conference on machine learning. PMLR, 2018.
>
> [4] Florina et.al, Ensemble Adversarial Training: Attacks and Defenses, ICLR (2018)
>
> > Q: One of the findings that DGCNN is robust has been discovered by prior arts.
>
> A: We disagree with the reviewer that the pointed paper has comprehensively studied the robustness of different architectures.
>
> First, the pointed paper [1] proposes a transferable attack targeting PointNet, PointNet++, and DGCNN models. The paper’s analysis is purely from the perspective of the proposed transferable attack. However, the transferable attack itself is a weaker attack since it is done in a gray-box setting. Existing work [2,3] and we have demonstrated that with a moderate with-box attack setting, the accuracy of DGCNN can be reduced to 0. Therefore, only analyzing a gray-box attack’s success rates on clean trained models cannot provide in-depth insights into different architectures’ adversarial robustness.
>
> Second, after a closer examination of [1], we find that the authors claimed to evaluate their attack against a naive adversarial training baseline in their appendix with the same setting in [4]. **However, [4] did not include anything about adversarial training.** Therefore, we cannot assess how the authors perform adversarial training to evaluate their attack.
>
> In comparison, **our analysis is much more comprehensive since we have validated our findings, including that DGCNN is more robust in multiple datasets (ModelNet40, ModelNet10, ScanObjectNN, and ShapeNetPart), threat models (point shifting, adding, and dropping), and tasks (classification and part segmentation)**. Second, we also conduct systematic analysis to connect self-supervised learning to the robustness of DGCNN, which is the key in our work to further contribute significant robustness enhancement. Furthermore, our analysis is based on a **formally defined adversarial training framework**. From a defensive perspective, we provide many more valuable insights for the community and motivate future research to design more robust model architectures and self-supervised learning tasks in 3D vision.
>
> [1] Hamdi, Abdullah, et al. "Advpc: Transferable adversarial perturbations on 3d point clouds." European Conference on Computer Vision. Springer, Cham, 2020.
>
> [2] Lee, Kibok, et al. "ShapeAdv: Generating Shape-Aware Adversarial 3D Point Clouds." arXiv preprint arXiv:2005.11626 (2020).
>
> [3] Wen, Yuxin, et al. "Geometry-aware generation of adversarial point clouds." IEEE Transactions on Pattern Analysis and Machine Intelligence (2020).
>
> [4] Xiang, Chong, Charles R. Qi, and Bo Li. "Generating 3d adversarial point clouds." Proceedings of the IEEE/CVF Conference on Computer Vision and Pattern Recognition. 2019.

---

> > ### Author Response · Authors · 2021-08-10
> > **Response to Reviewer ygsD [2/2]**
> >
> > > Q: The experimental setups might be flawed.
> >
> > A: We believe our experimental setups do make sense for the following reasons:
> >
> > 1. First, we would like to kindly highlight that we utilize $\epsilon = 0.05$ instead of 0.005, which is incorrectly indicated in the review.,
> >
> > 2. Additionally, we argue that $\epsilon = 0.05$ is already a very large perturbation. As Table 1 shows, our attack with $\epsilon = 0.05$ reduces all the clean trained models’ robust accuracy to 0 or near 0. We have also tested that even with $\epsilon = 0.02$, the adversary can still reduce all the models’ robust accuracy to less than 5%. On the other hand, the perturbed point clouds with $\epsilon = 0.05$ are at the edge of correct human predictions of objects. As we are not allowed to insert links, we kindly refer the reviewer to Figure 9 in [1], which also indicates the same conclusion. Numerically, $\epsilon = 0.05$ out of the range [-1,1] is also similar to the commonly used  $\epsilon = \frac{8}{255}$ in 2D adversarial training [2].
> >
> > 3. We use the experimental guidelines in [1] because it is the only existing work that leverages adversarial training analysis in 3D point clouds.
> >
> > We follow the reviewer’s suggestions to evaluate our adversarially trained model on more attacks including PGD attacks with $\epsilon = 0.02/0.04/0.06$ and C&W attack.
> >
> > The results of PDG attack with different  $\epsilon = 0.02/0.04/0.06$  as shown as follows.
> >
> > |                |     | ModelNet40  |                 | |       ScanObjectNN         |      | |   ModelNet10     |      |
> > |    :----:   |:----:   |:----:   |:----:   |:----:   |:----:   |:----:   |:----:   |:----:   |:----:   |
> > |           RA(%)           | PointNet   | DGCNN     | PCT       | PointNet     | DGCNN     | PCT       | PointNet   | DGCNN     | PCT       |
> > | AT Baseline ($\epsilon=0.02$)    | 63.2±0.33  | 80.9±0.22 | 78.8±0.15 | 43.6±0.31    | 54.9±0.55 | 54.1±0.33 | 91.9±0.21  | 93.3±0.19 | 92.1±0.13 |
> > | Best Finetuned ($\epsilon=0.02$) | 76.0±0.19  | 84.3±0.19 | 79.4±0.17 | 47.4±0.23    | 62.0±0.51 | 56.1±0.49 | 92.1±0.25  | 96.2±0.26 | 94.6±0.16 |
> > | AT Baseline ($\epsilon=0.04$)    | 46.2±0.3   | 70.9±0.3  | 61.3±0.11 | 28.5±0.19    | 39.6±0.45 | 35.1±0.21 | 84.5±0.36  | 91.0±0.37 | 86.2±0.29 |
> > | Best Finetuned ($\epsilon=0.04$) | 61.1±0.15  | 76.7±0.25 | 62.9±0.13 | 31.8±0.4     | 48.9±0.6  | 37.4±0.39 | 85.2±0.26  | 92.9±0.44 | 88.4±0.41 |
> > | AT Baseline ($\epsilon=0.06$)    | 25.1±0.51  | 48.9±0.69 | 37.2±0.33 | 17.8±0.15    | 25.6±0.33 | 19.1±0.4  | 74.1±0.12  | 84.2±0.55 | 77.0±0.29 |
> > | Best Finetuned ($\epsilon=0.06$) | 43.2±0.41  | 54.1±0.75 | 39.6±0.2  | 19.3±0.23    | 33.1±0.65 | 20.5±0.37 | 75.0±0.2   | 86.4±0.48 | 77.6±0.47 |
> >
> > From this table, we observe that our model could achieve non-trivial robustness among different epsilons (even with  $\epsilon > 0.05$), and it is expected that the adversarially trained models perform better with $\epsilon < 0.05$. Our model trained with self-supervised learning also achieves consistently stronger robustness than the AT and ST baselines. It further verifies the significance of self-supervised learning for 3D point cloud robustness.
> >
> > In the following table, we show the robustness of our best fine-tuned model against target C&W attack. For each test sample, we randomly select a class as a target. We also compare it with two baselines, including standard training (ST) baseline and adversarial training (AT) baseline. Note that the numbers here are the attack success rate since we use target attacks (the lower, the more robust). From the result, we could observe that our method still achieves consistently higher robustness compared to the two baseline methods.  It also verifies the significance of self-supervised learning for 3D point cloud robustness.
> >
> > |                |     | ModelNet40  |                 | |       ScanObjectNN         |      | |   ModelNet10     |      |
> > |    :----:   |:----:   |:----:   |:----:   |:----:   |:----:   |:----:   |:----:   |:----:   |:----:   |
> > | ASR(%)         | PointNet   | DGCNN | PCT  | PointNet     | DGCNN | PCT  | PointNet   | DGCNN | PCT  |
> > | ST Baseline    | 98.3       | 98.1  | 99   | 100          | 100   | 100  | 95.3       | 96.2  | 95.1 |
> > | AT Baseline    | 11.2       | 7.6   | 9.8  | 35.9         | 24.4  | 39.7 | 5.9        | 5.5   | 6    |
> > | Best Finetuned | 6.93       | 5.21  | 5.68 | 30.1         | 20    | 30.4 | 5.5        | 4.2   | 5.5  |
> >
> > [1] Sun, Jiachen, et al. "On Adversarial Robustness of 3D Point Cloud Classification under Adaptive Attacks." arXiv preprint arXiv:2011.11922 (2020).
> >
> > [2] Madry, Aleksander, et al. "Towards deep learning models resistant to adversarial attacks." arXiv preprint arXiv:1706.06083 (2017).
> >
> > [3] Athalye, Anish, Nicholas Carlini, and David Wagner. "Obfuscated gradients give a false sense of security: Circumventing defenses to adversarial examples." International conference on machine learning. PMLR, 2018.
> >
> > [4] Zhou, Hang, et al. "Dup-net: Denoiser and upsampler network for 3d adversarial point clouds defense." Proceedings of the IEEE/CVF International Conference on Computer Vision. 2019.
> >
> > [5] Ma, Chengcheng, et al. "Towards Effective Adversarial Attack Against 3D Point Cloud Classification." 2021 IEEE International Conference on Multimedia and Expo (ICME). IEEE, 2021.
> >
> > > Q: The uncertainty in the results is not quantified.
> >
> > A: We have included the uncertainty analysis in Section 3.2.3 and shown the standard deviation of the robust accuracy in Figure 3. Additionally, we have re-run 5 times using different random seeds for our best setting and included the average and variance in Appendix C.2.2. It is worth noting the variances of the models trained using our best settings are very small (less than 0.5%) among different architectures and datasets. It is really hard to report all experiments’ uncertainty since we have conducted very large-scale experiments.
> >
> > We will fix the grammar and presentation issues and cite the recent related works in our revised manuscript. Thanks again for the careful review.
> >
> > We believe that we have addressed all the raised concerns, and we are happy to answer any follow-up questions. We hope the reviewer might reconsider the rating based on our response.

---

> > > ### Comment · Reviewer_ygsD · 2021-08-12
> > > **good rebuttal**
> > >
> > > I would like to thank the authors for the excellent rebuttal. I will increase my score to 7, but I still have a couple of concerns.
> > >
> > > - Yes I agree about formalizing the adversarial training setup and conducting comprehensive evaluations. But the technical novelty is still an issue. I would assume that jointly performing SSL and AT should be a sound and novel contribution. However, the shown results validate that simply performing  SSL pretraining followed by AT is better.
> > >
> > > - The paper lacks expressive visualizations. Showing 400 numbers in a single table will not convey any information to the reader. This must be improved in the manuscript before it is published. I hope the authors take this point seriously.

---

> > > > ### Author Response · Authors · 2021-08-18
> > > > **Thank you for your decision!**
> > > >
> > > > We appreciate the reviewer for her/his valuable comments on our rebuttal and for agreeing to raise the score.
> > > >
> > > > We are glad that the reviewer agrees with several aspects of the novelty of our work. We also would like to highlight that we are the **first** to conduct such systematic analysis in 3D point cloud data. Therefore, as the reviewer indicated, we believe the overall framework and experimental results are meaningful for the community.
> > > >
> > > > Besides, the reason that adversarial pre-training for fine-tuning (APF) performs better than adversarial joint training (AJT) could also be attributed to the unique features in point cloud data. In our AJT experiments, the SSL task is to predict the transformation itself (i.e., rotation, patch permutation). **The spatial positions of the point cloud are totally different.** It is, therefore, hard to generalize the two distributions in our problem setting. We compute the mean and variance of clean data, rotated data, and displaced data (for jigsaw) on ModelNet40 to show the distributional gap from one perspective:
> > > >
> > > > |          | Original Data                           | Rotation $\eta=6$                       | Rotation $\eta=18$                     | Jigsaw k=3                 | Jigsaw k=4                 |
> > > > |----------|-----------------------------------------|-----------------------------------------|----------------------------------------|----------------------------|----------------------------|
> > > > | Mean  [x,y,z]   | [ 1.9201e-05, -8.8720e-05, -1.7456e-05] | [ 1.6767e-05, -6.3794e-05, -4.2133e-05] | [-1.1208e-05, -1.0000e-04, 1.1255e-04] | [ 0.0027, -0.0077, 0.0081] | [ 0.0045, 0.0053, -0.0047] |
> > > > | Variance  [x,y,z] | [0.1058, 0.1121, 0.1329]                | [0.1080, 0.1183, 0.1244]                | [0.1099, 0.1182, 0.1227]               | [0.3322, 0.3275, 0.3304]   | [0.3314, 0.3358, 0.3321]   |
> > > >
> > > > From the above table, we can see that displaced data (for jigsaw) have totally different distributions. Although the gap between rotated and original data is not as large as displaced data, it is also hard for the model to adapt to the rotated data since the architecture design is not invariant to rotation. Contrarily, APF is a more natural scheme for the backbones to learn robust context information. Since APF is done in sequential order, the backbones are able to learn the target distributional data smoothly while preserving robust prior features.
> > > >
> > > > We agree with the reviewer that we need to further polish the manuscript to incorporate concise and meaningful visualizations to help readers understand our study. We plan to use line/bar charts to plot the results in Tables 2 and 3. We will definitely take this factor seriously and address it before our work is published.

---

### Official Review · Reviewer_euZV · 2021-07-15

**Rating:** 6
**Confidence:** 5

**Summary:**

This paper systematically analyzes robustness of self-supervised adversarial training against adversarial attacks, which conducts thorough evaluation on several representative point cloud classifiers with different types of adversaries. Beyond point cloud classification, the task of part segmentation is also compared and analyzed.

**Limitations And Societal Impact:**

The following concerns can be important to further improve the paper:
1. There exists a large number of abbr. to make the paper very difficult to digest. Only the important and popular terms are suggested to use abbr.. More importantly, the definition of performance metrics CA and RA is missing in the paper. I suppose CA and RA could refer to clean accuracy and robust accuracy, but their forms remain unclear to the reviewer.

2. Technically, the threat models such as PS, PD, and PA and their experimental settings are important and sensitive to adversarial robustness. However, generation of adversaries should be well investigated and evaluated, which is missing in the experiments. For example, how existence of outliers affect adversarial robustness?

3. The APT performs better than AJT in experiments. The authors claim that distribution gap between SSL and original recognition tasks leads to poor classification performance of AJT. However, in UDA on point cloud classification, SSL is verified its effectiveness to mitigate domain gap (see [a]). Can you explain different performance of your paper and [a]?
[a] I.  Achituve, H. Maron, and G. Chechik. Self-supervised learning for domain adaptation on point cloud. CVPR 2021.

4. Insight given in the experiments of part segmentation encourages a more interesting SO(3)/SO(3) setting used to evaluate adversarial robustness of rotation-equivariant methods.


**Main Review:**

The paper reveals positive impact of adversarial training on improving adversarial robustness of multiple point cloud classifiers. I really appreciate that the authors conduct a thorough evaluation under different settings. The observation that robust local features are vital to achieve adversarial robustness is interesting and also meets our common knowledge.

**Time Spent Reviewing:**

8

---

> ### Author Response · Authors · 2021-08-10
> **Response to Reviewer euZV**
>
> We thank the reviewer for her/his constructive comments and agreeing that our experiments are thorough with useful insights. Below we respond to each question of the reviewer:
>
> > Q: There exists a large number of abbr. to make the paper very difficult to digest.
>
> A: We use a large number of abbr. due to the space constraints. We will follow the reviewer’s suggestions to revise our manuscript to reduce the number of abbr.to make them more clear. Yes, CA and RA mean clean accuracy and robust accuracy, respectively.
>
> > Q: The threat models and their experimental settings should be investigated and the generality of the attacks should be reviewed.
>
> A: In our study, we follow two principles to set up the threat model. First, the perturbation will not affect human perception much. Second, the perturbation is more than effective to break the clean trained models.
>
> Point shifting (PS) is a well-established attack in prior literature [1,2,3]. We follow the experimental setups in [3] in our study since [3] is the only one that also targets adversarial training-based methods. We believe this threat model does make sense. As Table 1 shows, our L-inf norm distance $\epsilon = 0.05$ reduces all the clean trained models’ robust accuracy to 0 or near 0. We have also tested that even with $\epsilon = 0.02$, the adversary can still reduce all the models’ robust accuracy to less than 5%. On the other hand, the perturbed point clouds with $\epsilon = 0.05$ are at the edge of correct human predictions of objects.
> As we are not allowed to insert links, we kindly refer the reviewer to Figure 9 in [3], which also indicates the same conclusion. Numerically, $\epsilon = 0.05$ out of [-1,1] is also similar to the commonly used  $\epsilon = \frac{8}{255}$ in 2D adversarial training [4].
>
> Unlike point shifting (PS) attacks, there is no defensive analysis point adding (PA) and dropping (PD) attacks since they are bounded by L-0 distance which is not differentiable. Adding or dropping a large amount of points will also have a major effect on human perception [1,5]. In the published paper that proposes PA attack [1], they suggest adding 50 to 100 points bounding by informal Chamfer distance. Therefore, we adopt their experimental setting to make sure that our threat model will not affect human perception. Since the attacker only has the ability to modify these newly added 100 points, it cannot reduce the accuracy to near 0. Similarly, the original paper that proposes PD has claimed dropping 200 points has great potential to even fool human perception [5]. Therefore, we restrict our threat model to be able to drop 100 points. Our evaluation results are also consistent with [5] in terms of both clean and robust accuracy.
>
> Therefore, we believe that our threat models are reasonable.
>
> For the generality of the threat models, we conduct an additional experiment using $\epsilon = 0.06$ which is larger than the adopted $\epsilon=0.05$. The results are shown as follows.
>
> |                |     | ModelNet40  |                 | |       ScanObjectNN         |      | |   ModelNet10     |      |
> |    :----:   |:----:   |:----:   |:----:   |:----:   |:----:   |:----:   |:----:   |:----:   |:----:   |
> |           RA(%)           | PointNet   | DGCNN     | PCT       | PointNet     | DGCNN     | PCT       | PointNet   | DGCNN     | PCT       |
> | ST Baseline ($\epsilon=0.06$)    | 0  | 0 | 0 | 0    | 0 | 0  | 0  | 0 | 0 |
> | AT Baseline ($\epsilon=0.06$)    | 25.1±0.51  | 48.9±0.69 | 37.2±0.33 | 17.8±0.15    | 25.6±0.33 | 19.1±0.4  | 74.1±0.12  | 84.2±0.55 | 77.0±0.29 |
> | Best Finetuned ($\epsilon=0.06$) | 43.2±0.41  | 54.1±0.75 | 39.6±0.2  | 19.3±0.23    | 33.1±0.65 | 20.5±0.37 | 75.0±0.2   | 86.4±0.48 | 77.6±0.47 |
>
> From this table, we observe that our model could achieve non-trivial robustness among different epsilons (even with  $\epsilon > 0.05$). Our model trained with self-supervised learning also achieves consistently stronger robustness than the AT and ST baselines. It further verifies the significance of self-supervised learning for 3D point cloud robustness.
>
> [1] Xiang, Chong, Charles R. Qi, and Bo Li. "Generating 3d adversarial point clouds." Proceedings of the IEEE/CVF Conference on Computer Vision and Pattern Recognition. 2019.
>
> [2] Liu, Daniel, Ronald Yu, and Hao Su. "Adversarial shape perturbations on 3D point clouds." European Conference on Computer Vision. Springer, Cham, 2020.
>
> [3] Sun, Jiachen, et al. "On Adversarial Robustness of 3D Point Cloud Classification under Adaptive Attacks." arXiv preprint arXiv:2011.11922 (2020).
>
> [4] Madry, Aleksander, et al. "Towards deep learning models resistant to adversarial attacks." arXiv preprint arXiv:1706.06083 (2017).
>
> [5] Zheng, Tianhang, et al. "Pointcloud saliency maps." Proceedings of the IEEE/CVF International Conference on Computer Vision. 2019.
>
> > Q: Why is there a distributional gap in adversarial joint training (AJT) between the classification and self-supervised learning tasks?
>
> A: We thank the reviewer for pointing out related work for us. There are two differences between our AJT and [1].
> First, the goal in our study is different from [1]. [1] tries to adapt the backbone to different styles of point clouds. However, our goal is to improve the adversarial robustness of the recognition task on the original distribution. Therefore, the generalization/adaptation to the rotation and jigsaw prediction task may distract the adversarial training on the recognition task in AJT.
> Second, the self-supervised learning (SSL) task in [1] is to reconstruct point clouds between two different styles. As shown in Figure 1 in [1], the point cloud between two domains still align well (i.e., no transformation applied to the point cloud) Therefore, we believe the distributional gap is not large between two styles of point clouds.
> In our AJT experiments, the SSL task is to predict the transformation itself (i.e., rotation, patch permutation). **The spatial positions of the point cloud are totally different.** It is also hard to generalize the two distributions in our problem setting. We compute the mean and variance of clean data, rotated data, and displaced data (for jigsaw) on ModelNet40 to show the distributional gap from one perspective:
>
> |          | Original Data                           | Rotation $\eta=6$                       | Rotation $\eta=18$                     | Jigsaw k=3                 | Jigsaw k=4                 |
> |----------|-----------------------------------------|-----------------------------------------|----------------------------------------|----------------------------|----------------------------|
> | Mean  [x,y,z]   | [ 1.9201e-05, -8.8720e-05, -1.7456e-05] | [ 1.6767e-05, -6.3794e-05, -4.2133e-05] | [-1.1208e-05, -1.0000e-04, 1.1255e-04] | [ 0.0027, -0.0077, 0.0081] | [ 0.0045, 0.0053, -0.0047] |
> | Variance  [x,y,z] | [0.1058, 0.1121, 0.1329]                | [0.1080, 0.1183, 0.1244]                | [0.1099, 0.1182, 0.1227]               | [0.3322, 0.3275, 0.3304]   | [0.3314, 0.3358, 0.3321]   |
>
> From the above table, we can see that displaced data (for jigsaw) have totally different distributions. Although the gap between rotated and original data is not as large as displaced data, it is also hard for the model to adapt to the rotated data since the architecture design is not invariant to rotation.
>
> To further eliminate the reviewer’s concern, we additionally conduct the experiments using the autoencoder task in AJT. The autoencoder task takes the same point cloud as the supervised classification task, so that the distributional gap is relatively small. The following table shows the results.
>
> |                |     | ModelNet40  |                 | |       ScanObjectNN         |      | |   ModelNet10     |      |
> |    :----:   |:----:   |:----:   |:----:   |:----:   |:----:   |:----:   |:----:   |:----:   |:----:   |
> | CA/RA (%) | PointNet   | DGCNN     | PCT       | PointNet     | DGCNN     | PCT       | PointNet   | DGCNN     | PCT       |
> |   sphere  | 87.5/44.4  | 90.9/62.1 | 89.6/49.2 | 68.9/24.2    | 75.5/36.5 | 72.5/20.5 | 96.7/79.8  | 98.2/86.3 | 97.5/80.3 |
> |   plane   | 87.4/42.1  | 90.7/61.9 | 89.3/48.7 | 68.5/23.9    | 75.6/34.7 | 72.8/20.6 | 96.7/79.7  | 98.1/86.2 | 97.4/79.9 |
> |  gaussian | 86.9/43.9  | 90.9/61.9 | 88.9/49.2 | 68.7/24.4    | 76.3/35.1 | 72.1/20.5 | 96.6/79.7  | 98.3/86.9 | 97.5/80.0 |
>
> We find that the results of AJT using the autoencoder task are more stable than the other two tasks. We believe it is because the input for the autoencoder is the same as the recognition task so that the distributional gap is small. However, it is still worse than our pre-training scheme. We will add this experiment to our manuscript and discuss the results.
>
> > Q: Insight given in the experiments of part segmentation encourages a more interesting SO(3)/SO(3) setting used to evaluate adversarial robustness of rotation-equivariant methods.
>
> A: We agree with the reviewer’s assessment. We do not evaluate the robustness of rotation-equivariant methods since (1) existing rotation-equivariant models on point clouds cannot achieve state-of-the-art performance on the classification and part segmentation tasks [1,2] on the clean dataset and (2) the current scope of our study is within the L-p norm threat model (L-p based adversarial examples).  We leave this SO(3)/SO(3) setting as our future work and will discuss it in our manuscript.
>
> [1] https://paperswithcode.com/task/3d-part-segmentation/latest
>
> [2] https://paperswithcode.com/task/3d-point-cloud-classification
>
> We believe that we have addressed all the raised concerns, and we are happy to answer any follow-up questions. We hope the reviewer might reconsider the rating based on our response.

---

> > ### Comment · Reviewer_euZV · 2021-09-03
> > **Response**
> >
> > Thanks for the efforts spent on the rebuttal, but most of my concern has been addressed. As a result, I would like to raise my score to ``6''.

---

> > > ### Author Response · Authors · 2021-09-03
> > > **Thank you for your decision!**
> > >
> > > We are glad to hear that our response addressed most of the concerns from the reviewer. Your comments are essential for us to improve our manuscript. Thank you!

---

### Decision · Program_Chairs · 2021-09-28

**Decision:**

Accept (Poster)

**Comment:**

I recommend acceptance. This is solid experimental work on an interesting and relevant question: the adversarial robustness properties of the most popular 3D point cloud architectures. The question is relevant as a lot of follow up work and engineering efforts are built on these and similar architectures. Even though I accept some reviewer’s concerns that the paper might be thin on novelty, my opinion is that this solid experimental study and its findings will be of interest to the community.

**Consistency Experiment:**

NeurIPS has a long history of experimentation. In 2014, NeurIPS ran an experiment in which 10% of submissions were reviewed by two independent committees to quantify the randomness in the review process. This year, we repeated a variant of this experiment to see how the quality of the review process has changed over time.  This paper was part of the experiment and was therefore assigned to two committees (consisting of reviewers, an Area Chair, and a Senior Area Chair) that reached independent decisions.  If both committees made the same recommendation, this recommendation was followed. If a single committee recommended acceptance, the paper was accepted (with the exception of a few cases in which the other committee identified what we considered a fatal flaw, e.g., an error in a key result).

Both committees reached the same decision: **Accept (Poster)**

The other committee assigned to the paper recommended **Accept (Poster)**.  You can find the other set of reviews, along with any follow up discussion with the authors here:
https://openreview.net/forum?id=srHp6A1c2z-